# *Salmonella* manipulates macrophage migration via SteC-mediated myosin light chain activation to penetrate the gut-vascular barrier

Yuanji Dai [ID] [1,2,9], Min Zhang [ID] [1,2,9], Xiaoyu Liu [ID] [1,2,9], Ting Sun[3,4,9], Wenqi Qi[2], Wei Ding[5], Zhe Chen [ID] [6], Ping Zhang [ID] [1,2], Ruirui Liu[1,2], Huimin Chen[3], Siyan Chen[2], Yuzhen Wang[1,2], Yingying Yue[1,2], Nannan Song[1,2], Weiwei Wang[1,2], Haihong Jia[1,2], Zhongrui Ma[2,3], Cuiling Li[1,2], Qixin Chen [ID] [3✉] & Bingqing Li [ID] [1,2,3,7,8✉]

## Abstract

The intestinal pathogen *Salmonella enterica* rapidly enters the bloodstream after the invasion of intestinal epithelial cells, but how *Salmonella* breaks through the gut-vascular barrier is largely unknown. Here, we report that *Salmonella* enters the bloodstream through intestinal CX3CR1[+] macrophages during early infection. Mechanistically, *Salmonella* induces the migration/invasion properties of macrophages in a manner dependent on host cell actin and on the pathogen effector SteC. SteC recruits host myosin light chain protein Myl12a and phosphorylates its Ser19 and Thr20 residues. Myl12a phosphorylation results in actin rearrangement, and enhanced migration and invasion of macrophages. SteC is able to utilize a wide range of NTPs other than ATP to phosphorylate Myl12a. We further solved the crystal structure of SteC, which suggests an atypical dimerization-mediated catalytic mechanism. Finally, in vivo data show that SteC-mediated cytoskeleton manipulation is crucial for *Salmonella* breaching the gut vascular barrier and spreading to target organs.

Keywords *Salmonella* Effector; Kinase; Host–Microbe Interaction; Cytoskeleton; Macrophage Migration
Subject Categories Cell Adhesion, Polarity & Cytoskeleton; Microbiology, Virology & Host Pathogen Interaction

## Introduction

*Salmonella* is a significant pathogen that causes food poisoning and typhoid fever in both animals and humans (Dougan and Baker, 2014; Marchello et al, 2022; Ten Hoeve et al, 2022). *Salmonella*

enters the human body after consumption of contaminated food or water, invades the epithelial cells of the small intestine, and then spreads to target tissues (Santos et al, 2001). A hematogenous route via CD18[+] phagocytes is the primary mode of travel for *Salmonella* from the gastrointestinal tract to the liver and spleen (Kim et al, 2017; Vazquez-Torres et al, 1999; Watson and Holden, 2010). Intriguingly, the gut-vascular barrier in the human intestine acts to prevent the transmission of pathogenic bacteria, but *Salmonella* can rapidly break through this barrier and spread to the liver and blood in a *Salmonella* pathogenicity island-2 (SPI-2)-dependent manner (Bertocchi et al, 2021; Spadoni et al, 2015). However, the exact mechanism of this process remains largely unclear.

*Salmonella* mainly parasitizes in macrophages of the target organs (Garcia-del Portillo, 2001; Nix et al, 2007; Richter-Dahlfors et al, 1997). When *Salmonella* survives in macrophages, the pathogenicity island-2 type III secretion system (SPI-2 T3SS) can inject as many as 28 effectors into the host cytoplasm (Jennings et al, 2017; McGhie et al, 2009). SteC is one of the SPI-2-encoded effectors and the only kinase effector of *Salmonella* (Geddes et al, 2005; Poh et al, 2008). SteC displays significantly low homology with all eukaryotic and prokaryotic kinases and there are no structures similar to SteC (with greater than 30% identity) in the PDB database. SteC can induce actin rearrangement in fibroblasts partly by activating the phosphorylation of MEK (Odendall et al, 2012). Other host substrates of SteC, such as Hsp27 and FMNL, were identified via large-scale screening (Fernandez-Pinar et al, 2012; Imami et al, 2013; Walch et al, 2021). However, few studies have investigated the function and detailed mechanism of SteC when *Salmonella* survives in macrophages, the natural host of *Salmonella*.

In this work, we discovered that within 1 h after infection, *Salmonella* have been observed to accumulate in CX3CR1[+] macrophages that surround the microvessels in the villi of the

[1]Department of Clinical Laboratory, Shandong Provincial Hospital Affiliated to Shandong First Medical University, Jinan 250021, China. [2]Department of Pathogen Biology, School of Clinical and Basic Medical Sciences, Shandong First Medical University & Shandong Academy of Medical Sciences, Jinan, China. [3]School of Pharmaceutical Sciences, Medical Science and Technology Innovation Center, Shandong First Medical University & Shandong Academy of Medical Sciences, Jinan 250062, China. [4]School of Pharmaceutical Sciences, Cheeloo College of Medicine, Shandong University, Jinan 250012, China. [5]Beijing National Laboratory for Condensed Matter Physics, Institute of Physics, Chinese Academy of Sciences, Beijing 100190, China. [6]State Key Laboratory of Microbial Technology, Shandong University, Qingdao 266237, China. [7]Key Lab for Biotech-Drugs of National Health Commission, Jinan 250117, China. [8]Key Lab for Rare & Uncommon Diseases of Shandong Province, Jinan 250117, China. [9]These authors contributed equally: Yuanji Dai, Min Zhang, Xiaoyu Liu, Ting Sun. ✉E-mail: chenqixin@sdfmu.edu.cn; bingqingsdu@163.com

small intestine. Moreover, *Salmonella*-infected macrophages have the ability to penetrate the bloodstream directly. Different from antigen-activated macrophages that are immobile, *Salmonella*-infected macrophages showed significantly increased migration/invasion properties, and this process depends on the host actin cytoskeleton and *Salmonella* effector SteC. Additionally, we have further discovered that SteC directly phosphorylates and activates the myosin light chain protein Myl12a in macrophage, independent of host kinases such as MLCK and ROCK. More interestingly, SteC can utilize four different NTPs as phosphate donors to overcome ATP competition with host kinase. The crystal structure of SteC revealed a unique dimerization-mediated kinase catalytic mechanism. The conserved Asp364 plays a crucial role in kinase catalysis, although this site was not previously predicted to be a key catalytic site based on sequence alignment (Heggie et al, 2021; Poh et al, 2008). Further, we found that a long helix in front of the kinase domain mimics the host Myl12a-interacting protein, which is crucial for Myl12a recruitment. Finally, in vivo experiments showed that SteC barely affects *Salmonella* proliferation in macrophages, but is crucial for *Salmonella* to break through the gut-vascular barrier and spread to target organs during infection.

# Results

## Intestinal CX3CR1+ macrophages carrying *Salmonella* enters the bloodstream in the early period of infection

Previous studies have shown that *Salmonella* spreads to target organs such as the liver and spleen through a lymphatic pathway mediated by M cells or DCs and blood pathway mediated by CD18-positive phagocytes (Jones et al, 1994; Rescigno et al, 2001; Vazquez-Torres et al, 1999). However, increasing evidence suggests that the blood pathway may be more crucial, as removal of MLN or CCR7 does not affect *Salmonella* dissemination to target organs (Carden et al, 2017; Kim et al, 2017; Lim et al, 2014; Voedisch et al, 2009; Watson and Holden, 2010). The precise cell population that carry *Salmonella* into the bloodstream has not been fully characterized (Kim et al, 2017; Vazquez-Torres et al, 1999). To investigate the mechanism by which *Salmonella* efficiently enters the bloodstream, C57BL/6J mice were orally infected with GFP-labeled *Salmonella*. Following oral infection for 0.5 and 1 h, the localization of *Salmonella* within the small intestine of infected mice was observed using immunohistochemistry and peripheral blood of infected mice was collected and further analyzed. After a 30-min period of oral infection, *Salmonella* were predominantly found within the epithelial cells of the small intestine. Additionally, a small amount of *Salmonella* (~15%) were observed around the microvasculature of the small intestine (Fig. 1A,C). After 1 h of oral infection, ~50% of *Salmonella* was observed surrounding the microvasculature of the intestine (Fig. 1B,C). Previous studies have indicated that one subset of macrophages, characterized by CX3CR1 positivity, is present along the vascular wall of the small intestine, forming an intestinal vascular barrier to prevent bacterial dissemination into the bloodstream (Honda et al, 2020). CX3CR1+ macrophages were also the major cell population harboring intracellular *Salmonella* shortly after infection (Man et al, 2017; Regoli et al, 2017). In this study, we performed immunofluorescence staining on the infected small intestine, confirming that

*Salmonella* predominantly localizes in macrophages around the microvasculature during the initial phases of infection (Fig. 1D). Furthermore, the bacterial load in the blood of infected mice was quantified. Initially, only a minimal number of bacteria were detected 0.5 h post-infection, while a substantial quantity of *Salmonella* infiltrated the bloodstream 1 h post-infection. Around $245.8 \pm 48.7$ *Salmonella* colonies were detected from 100 μL peripheral blood per mouse 1 h post-infection (Fig. 1E). To further determine the precise cell population containing GFP-labeled *Salmonella*, flow cytometric of peripheral blood leukocytes was performed. Around $252.5 \pm 68.3$ leukocyte cells carrying GFP-labeled *Salmonella* were observed in per 10,000 peripheral-blood leukocytes of infected C57BL/6 mice, whereas the control group only detected $138.8 \pm 30.4$ GFP+ leukocyte cells, which may represent the basal autofluorescence (Fig. 1F,G). Further analysis demonstrated that ~30% of GFP+ cells in the *Salmonella*-infected group were F4/80-positive, while only ~3% were CD11c-positive (Fig. 1H,I). Nearly all the GFP+ cells in control groups were both CD11c and F4/80 negative, suggesting a higher F4/80-positive ratio (60–70%) in GFP+ cells of *Salmonella*-infected group when false GFP positive cells caused by spontaneous fluorescence were deducted. The expression of CX3CR1 in GFP+F4/80+ cells was detected, roughly 30% of these cells co-expressed CX3CR1 (Fig. 1J,K). Immunofluorescence analysis of peripheral blood leukocytes provided additional evidence that *Salmonella* exists in F4/80-positive macrophages (Fig. 1L). The above data suggest that *Salmonella* directly enters the bloodstream through intestinal resident macrophages during early infection. However, it is not conventionally believed to occur that mature macrophages could return to the bloodstream.

## The migratory and invasiveness properties of macrophages are significantly increased during *Salmonella* infection in an actin- and SteC-dependent manner

During our previous research on *Salmonella*-infected macrophages, we observed that infected cells exhibited an enhanced motility compared with that of uninfected ones (Appendix Fig. S1; Movie EV1). Based on this, we speculated that *Salmonella* could activate the invasiveness of macrophages. To test this hypothesis, we characterized the migration and invasiveness of RAW264.7 macrophages before or after *Salmonella* challenge using a Boyden Chamber installation. The uninfected RAW264.7 cells exhibited weak migration or invasion, with significantly enhanced mobility and invasiveness after *Salmonella* infection (Fig. 2A,B). LPS and heat-treated *Salmonella* did not activate the motility behavior of macrophages, indicating that only live *Salmonella* could impel macrophages to move (Fig. 2A,B). To assess the extent to which *Salmonella* can enhance macrophage migration compared to the host cytokines, CCL2 and CCL7 were used as positive controls in this study. It was observed that the addition of CCL2 significantly enhanced macrophage mobility and invasion, whereas CCL7 had minimal effect. When comparing these findings to the effects of *Salmonella* infection on motility and invasion, we discovered that *Salmonella* infection induces a similar promotion of motility and invasion as CCL2 (Fig. 2C). Given the critical role of the cytoskeleton in cell migration, cytoskeletal inhibitors were used to assess the role of the cytoskeleton in *Salmonella*-induced

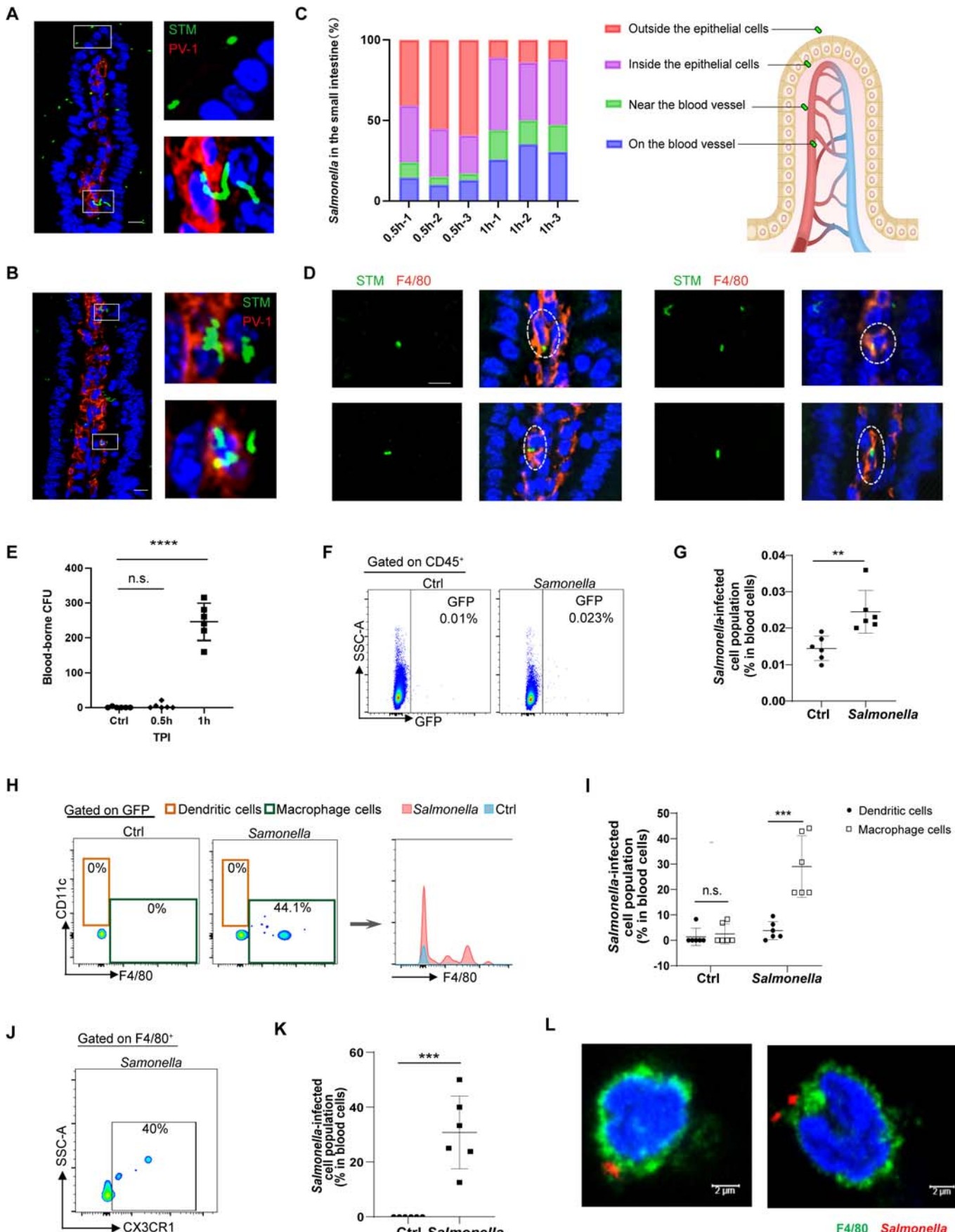

◀ **Figure 1. Intestinal CX3CR1⁺ macrophages carrying *Salmonella* enters the bloodstream in the early period of infection.**

C57BL/6 mice were injected orogastrically with PBS or $2 \times 10^8$ CFU per mouse of *S. Typhimurium* strain expressing GFP, tissue organs, and blood samples were collected at 30 min or 1 h post-infection for subsequent experiments. (**A, B**) The small intestinal of infected mice were analyzed through immunofluorescence. Small intestinal (SI) after 0.5 h (**A**) and 1 h (**B**) of *Salmonella* infection were stained with PV-1 (Red), DAPI (Blue), and *Salmonella* (Green). Scale bars, 10 μm. (**C**) Statistics of *Salmonella* distribution in host small intestinal at 0.5 h or 1 h after infection. $n = 3$ per group. (**D**) *Salmonella* primarily colonizes macrophages located in the vicinity of blood vessels. Small intestinal (SI) after 1 h of *Salmonella* infection were stained with F4/80 (Red), DAPI (Blue), and *Salmonella* (Green). The white dashed line indicates F4/80-positive macrophages containing bacteria. Scale bars, 10 μm. (**E**) The number of bacteria recovered from 100 μL peripheral blood per mouse. $n = 6$ per mouse. Data were mean ± SD. An unpaired *t*-test was used to determine the statistical significance between the two groups. ****$P < 0.0001$. (**F, G**) The white blood cells were isolated from infected C57BL/6 mice, and the GFP-positve cells were analyzed by flow cytometry 1 h post-infection. $n = 6$ per group. Data were mean ± SD. An unpaired *t*-test was used to determine the statistical significance between the two groups. **$P < 0.01$. (**H, I**) Representative flow cytometric analysis (**H**) and percentage (**I**) of dendritic cells (CD11c⁺F4/80⁻) and macrophage cells (F4/80⁺) in GFP-positve population from white blood cells from six mice per group. Data were mean ± SD. An unpaired *t*-test was used to determine the statistical significance between the two groups. ***$P < 0.001$. (**J, K**) Representative flow cytometric analysis (**J**) and percentage (**K**) of intestinal macrophage cells (CX3CR1⁺F4/80⁺) in GFP-positve population. $n = 6$ per group. Data were mean ± SD. An unpaired *t*-test was used to determine the statistical significance between the two groups. ***$P < 0.001$. Each dot indicates an individual animal. (**L**) Representative images of blood cells containing *Salmonella* (Red) associated with F4/80-expressing macrophage cells (Green), determined by confocal microscopy. Scale bars, 2 μm. Source data are available online for this figure.

macrophage migration. The results demonstrated that this process was dependent on actin, but not on tubulin (Fig. 2C).

When *Salmonella* survives in macrophages, it manipulates host cells by secreting SPI-2 effectors. Of these factors, SteC and SseI (also known as SrfH) are related to actin rearrangement (Poh et al, 2008; Worley et al, 2006). Knockout (KO) strains of *steC*, *sseI*, and *ssaW* (previously shown to be SPI-2 deficient) were constructed, and the mobility and invasiveness behavior of macrophages infected with these KO strains were detected. The results showed that the Δ*steC* and Δ*ssaW* strains completely lost the ability to induce cell migration while the Δ*sseI* strain exhibited partial ability to promote macrophage migration and invasiveness (Fig. 2D). The migration ability of Δ*steC*-infected macrophages was almost identical to that of uninfected cells and significantly lower than that of WT *Salmonella* (Movies EV1, 2), confirming that SteC is a key determinant of the enhanced mobility of macrophages induced by *Salmonella*.

A series of experiments were conducted using primary bone marrow-derived macrophages (BMDM) to evaluate their motility and invasiveness subsequent to *Salmonella* infection. *Salmonella* infection led to a significant increase in both motility and invasiveness in BMDM cells (Fig. 2E,F). However, the BMDMs infected with the Δ*steC* strain did not display an invasive phenotype, providing additional evidence that the migration and invasion properties of macrophages driven by *Salmonella* are dependent on SteC (Fig. 2E,F).

Previous studies have reported that *Salmonella* SteC can induce an actin-rearrangement phenotype in fibroblasts (Odendall et al, 2012). To test the situation in macrophages, the F-actin of macrophages were observed 8 h post-infection with WT or Δ*steC* *Salmonella*. The results revealed that WT infection could induce actin reorganization in macrophages, while infection with the Δ*steC* strain did not result in actin reorganization. Morphologically, the WT-infected macrophages are more spreading and adapt their migration mode, which were differed from the Δ*steC*-infected and uninfected ones (Fig. 2G,H).

## SteC induces actin rearrangement and the migration/ invasion properties of macrophages in a MEK/ERK/ MLCK pathway-independent manner

In order to better investigate the function of SteC in macrophages, a stable SteC-expressing RAW264.7 cell line (pSteC), a SteC K256H-expressing cell line (pSteC K256H, a mutant lacking kinase activity) (Odendall et al, 2012) and a GFP control cell line (pGFP) were established. The SteC and SteC K256H proteins are expressed at equivalent levels in the pSteC and pSteC K256H cell lines (Fig. 3A). No changes in proliferation were observed in cells with pSteC and pGFP. However, pSteC, but not pSteC K256H, displayed extremely enhanced migration and invasion (Fig. 3B–D), indicating that SteC directly activates the mobility and invasiveness of macrophages, and this activation requires its kinase activity. Actin rearrangement occurred in the SteC-expressing macrophages, and this was largely dependent on its kinase activity (Fig. 3E,F).

In 2012, it was reported that SteC activates actin rearrangement in fibroblasts by activating the MEK/ERK/MLCK/Myosin II pathway through phosphorylation on MEK (Odendall et al, 2012). However, several studies have reported that ERK phosphorylation in macrophages is significantly inhibited after *Salmonella* infection (Anand et al, 2012; Behnsen et al, 2015; Zaki et al, 2014), indicating that the SteC-MAPK-ERK activation pathway may not be applicable to macrophages. To investigate the role of the MAPK-ERK pathway in SteC-induced macrophage migration, MEK, MLCK, and ROCK inhibitors were added to SteC-expressing RAW264.7 cells, and relevant phenotypes were detected (Fig. 3G–J). Treatment with inhibitors of host ERK, MLCK, and ROCK showed almost no effect on the aforementioned phenotypes, indicating that SteC triggers actin rearrangement and cell migration of macrophages in a manner that is independent of the MEK-ERK-MLCK pathway.

## The macrophage interactor for SteC was identified as myosin light chain protein Myl12a

To identify the target of SteC in macrophages, we constructed a cDNA library of RAW264.7 macrophages and then SteC was used as a bait protein for yeast two-hybrid screening of the above cDNA library. A total of 46 positive clones were screened, of which 14 were sequenced and found to be myosin light chain protein Myl12a (Fig. 4A). The direct interaction between SteC and Myl12a was confirmed by yeast two-hybrid one-on-one verification (Fig. 4B). The actin-myosin cytoskeleton provides the force required for both cell motility and changes in morphology, and myosin light chain (MLC) is an important component of myosin (Vicente-Manzanares et al, 2009; Wilson et al, 2010). Importantly, the phosphorylation of MLC is the key step to induce actin rearrangement (Hirano and

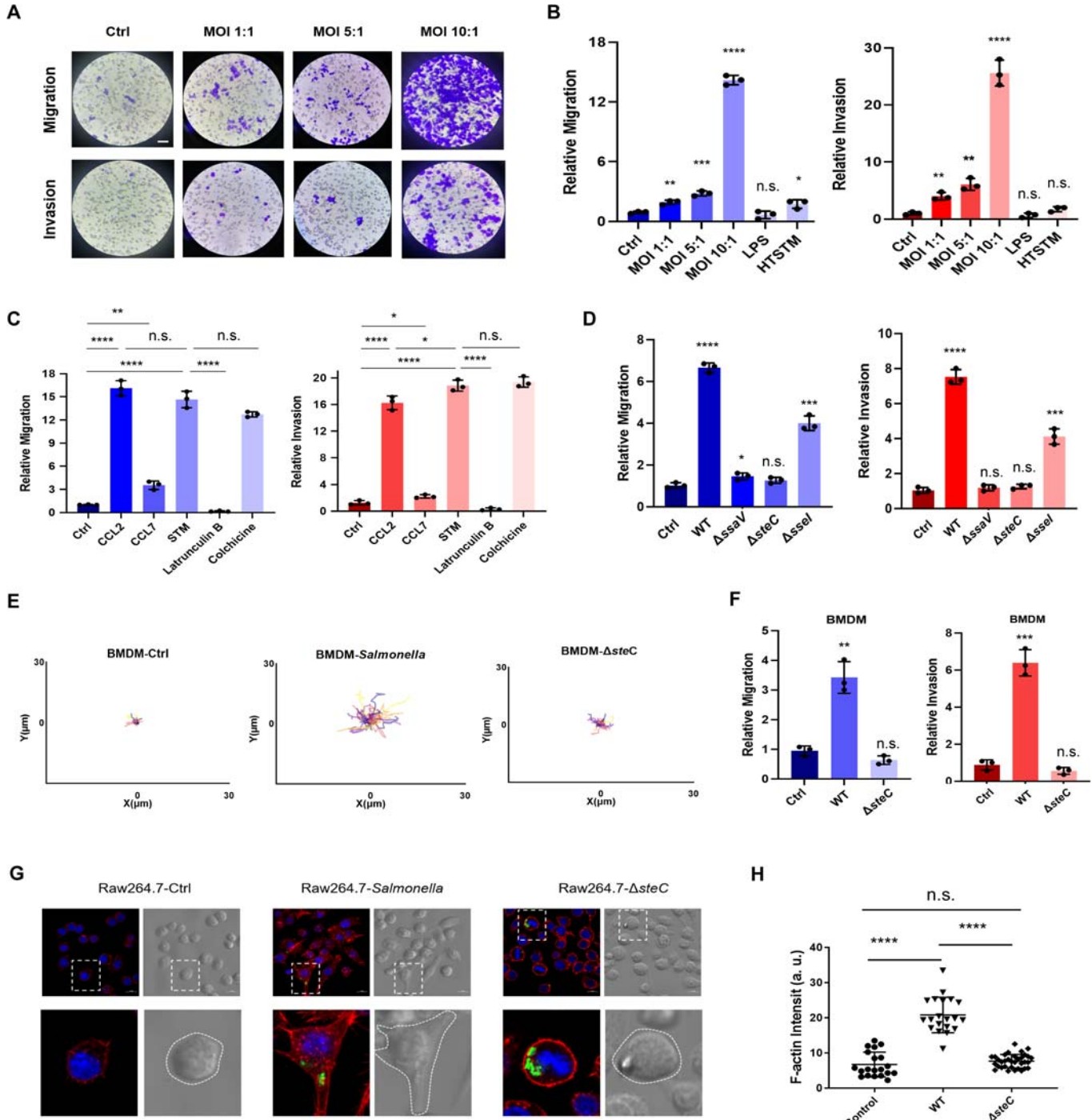

Hirano, 2016; Kohama et al, 1996; Park et al, 2011; Scholey et al, 1981). Thus, we speculated that the interaction between SteC and Myl12a is crucial for SteC to regulate cell motility. To test this, we further investigated the interaction between SteC and Myl12a by transfecting HeLa cells with fluorescently labeled SteC and Myl12a. The resulting fluorescence colocalization results provided additional validation of an in vivo association between SteC and Myl12a (Fig. 4C–E). Next, SteC, SteC-C (194 aa–457 aa, a fragment with kinase activity) (Odendall et al, 2012), and Myl12a proteins were

expressed in *E.coli* BL21 strain and purified for in vitro study. Pulldown assays clearly showed stable interactions between both SteC and SteC-C domains and Myl12a (Fig. 4F). To test this interaction further, biolayer interferometry assays (BLI) were used to quantify the strength of the interaction between SteC and Myl12a. The results showed the dissociation constant $K_d$ value of full-lengh SteC and Myl12a was $1.34 \pm 0.054 \, \mu M$ and that of the SteC-C fragment and Myl12a was $23.3 \pm 1.4 \, \mu M$ (Fig. 4G,H), a greater than tenfold reduction in binding.

**Figure 2.** *Salmonella* infection induces the migration and invasion properties of macrophages in an actin- and SteC-dependent manner.

(A, B) The migration and invasion of RAW264.7 cells infected by *Salmonella* were detected at the indicated MOI for 24 h. The cells that migrated to the lower chamber were stained and counted under light microscopy. Scale bars, 50 μm. The uninfected RAW264.7 cells were used as negative controls. LPS (1 μg/ml) and heat-treated *Salmonella* (95 °C 10 min, MOI 10:1) were also used as controls. (C) The migration and invasion of RAW264.7 cells treated by CCL2 (10 ng/ml) or CCL7 (100 ng/ml) or challenged by wild-type *Salmonella* (MOI 10:1) with or without colchicine (50 μM) or Latrunculin B (50 μM). (D) The migration and invasion of RAW264.7 cells were challenged by wild-type, ΔssaW, ΔsteC, or Δssel. MOI = 10. (E) The actual movement trajectory of bone marrow-derived macrophages (BMDM) is challenged by wild-type or ΔsteC. The uninfected BMDM cells were used as controls. More than 30 individual cells for each group were calculated. (F) The migration and invasion of BMDM cells infected by wild-type or ΔsteC were detected for 4 h. The uninfected BMDM cells were used as controls. (G) RAW264.7 cells infected by wild-type or ΔsteC expressing GFP (green) were stained for F-actin (red) and DAPI (blue). Scale bar, 2 μm. (H) The mean fluorescence intensity of intracellular F-actin related to H was quantified. More than 20 individual cells for each group were calculated. Data information: For 2B, 2C, 2D, and 2F, the amounts of migration and invasion of uninfected RAW264.7 cells or BMDM cells were set as 1. The mean ± SD of three biological replicates was shown. For 2H, The normalization of actin intensity is determined through the division of fluorescence intensity by the area. An unpaired *t*-test was used to determine the statistical significance between the two groups. The mean ± SD of 20 individual cells was shown. Unless otherwise specified, the infected group was compared with the uninfected control group. *$P < 0.05$, **$P < 0.01$, ***$P < 0.001$, ****$P < 0.0001$, n.s. not significant. Source data are available online for this figure.

## SteC activates Myl12a through direct phosphorylation of T19 and S20

SteC was previously demonstrated to be a kinase (Odendall et al, 2012), so we speculated that SteC may phosphorylate Myl12a. To test this, an in vitro phosphorylation experiment was performed with purified Myl12a, SteC, and SteC-C proteins using P$^{32}$ radioisotope-labeled ATP and Phos-Tag assay. The results demonstrate that SteC can phosphorylate Myl12a in vitro (Fig. 5A,B). Mass spectrometry results revealed that the molecular weight of SteC-modified Myl12a increased by 160 Da compared to that of native Myl12a, and further secondary mass spectrometry showed that T19 and S20 are the phosphorylation sites (Fig. 5C,D). T19 and S20 are also the key sites for MLC activation, and are the sites of phosphorylation by MLCK and ROCK kinases (Sebbagh et al, 2001; Totsukawa et al, 2000). Using phosphorylation antibodies for Myl12a T19 S20, we further confirmed that SteC catalyzes phosphorylation at both T19 and S20 of Myl12a (Fig. 5E). The Myl12a T19A, S20A double mutant was not phosphorylated by SteC confirming that T19 and S20 are the only two phosphorylation sites (Fig. 5F).

To assess whether SteC-mediated Myl12a phosphorylation occurred during infection, a series of *Salmonella* infection experiments were conducted using RAW264.7 cells and BMDM cells. The results showed that phosphorylation of Myl12a was dramatically upregulated in WT-infected cells 1 h post-infection, whereas no phosphorylation of Myl12a was detected in the ΔsteC-infected cells (Fig. 5G,H). Moreover, the phosphorylation of Myl12a was detected in BMDM cells as early as 15 min after wild-type *Salmonella* infection (Fig. 5I). Relevant experiments were conducted using SteC-expressing cell lines. The phosphorylation of T19 and S20 of Myl12a in SteC-expressing cells is significantly higher than in the K256H mutant and GFP-expressing strains (Fig. 5J). To further explore the potential substrates for SteC, phosphoproteomics were performed in pSteC cells and pGFP cells (Fig. 5K). The phosphorylation levels of various proteins in pSteC RAW264.7 were significantly enhanced, and Myl12a was identified as one of the top five proteins showing the highest fold increase (Fig. 5L; Appendix Tab. S1). The above data strongly support that SteC catalyzes the phosphorylation of Myl12a during *Salmonella* infection of macrophages.

Two homologous proteins, Myl12b and Myl9, are highly similar to Myl12a (97.66% and 94.15% identities, respectively) and are also expressed in macrophages (Appendix Fig. S2A). We next examined whether SteC could also bind to Myl12b or Myl9 and found that SteC can directly interact, phosphorylate, and activate Myl12b and Myl9 (Appendix Fig. S2B–D).

To determine the role of Myl12a in SteC-mediated motility of macrophage, a Myl12a knockout derivative of RAW264.7 was constructed, and then the mobility and invasion abilities of normal cells and knockout cells after *Salmonella* infection were observed (Appendix Fig. S3). The results reveal a decrease of 1/2 in mobility and invasiveness in the Myl12a knockout cells compared to the WT RAW264.7, indicating that Myl12a plays an important role in these processes. SteC could activate Myl12b and Myl9 as well as Myl12a, which may explain why the Myl12a knockout cells did not completely lose mobility and invasiveness.

## SteC is a distinctive kinase adopting four NTPs as phosphate donors

SteC displays significantly low homology with other eukaryotic and prokaryotic kinases. We investigated SteC kinase activity using *in vitro* phosphorylation experiments with different metal ions and four types of NTPs. Unexpectedly, SteC was able to phosphorylate Myl12a using all four types of NTPs as phosphate donors (Appendix Fig. S4A,B). To our knowledge, this is the only kinase with this characteristic. Then the kinetic analysis of SteC kinase using various NTPs was performed. The results showed that SteC phosphorylates Myl12a using four types of NTPs at similar affinity in the presence of Mn$^{2+}$, with $K_m$ values of 1.56 μM for ATP, 5.53 μM for UTP, and 5.84 μM for GTP (Appendix Fig. S4C,D).

## Structural insights into a dimerization-mediated catalytic mechanism of SteC kinase

No homologous structure of SteC was found in the PDB database, so the trRosetta server was used for SteC protein structure prediction (Du et al, 2021). According to the predicted structure, SteC has four parts: N domain, M helix, Kinase domain, and CC domain (Fig. 6A). Based on the predicted structure, we constructed six additional SteC fragments (202 aa–457 aa, 202 aa–375 aa, 168 aa–364 aa, 194 aa–364 aa, 202 aa–364 aa, 211 aa–364 aa) were constructed and their kinase activities were measured. The results indicate that in addition to the kinase domain, the CC domain is also crucial for the catalysis of Myl12a phosphorylation by SteC (Fig. 6B).

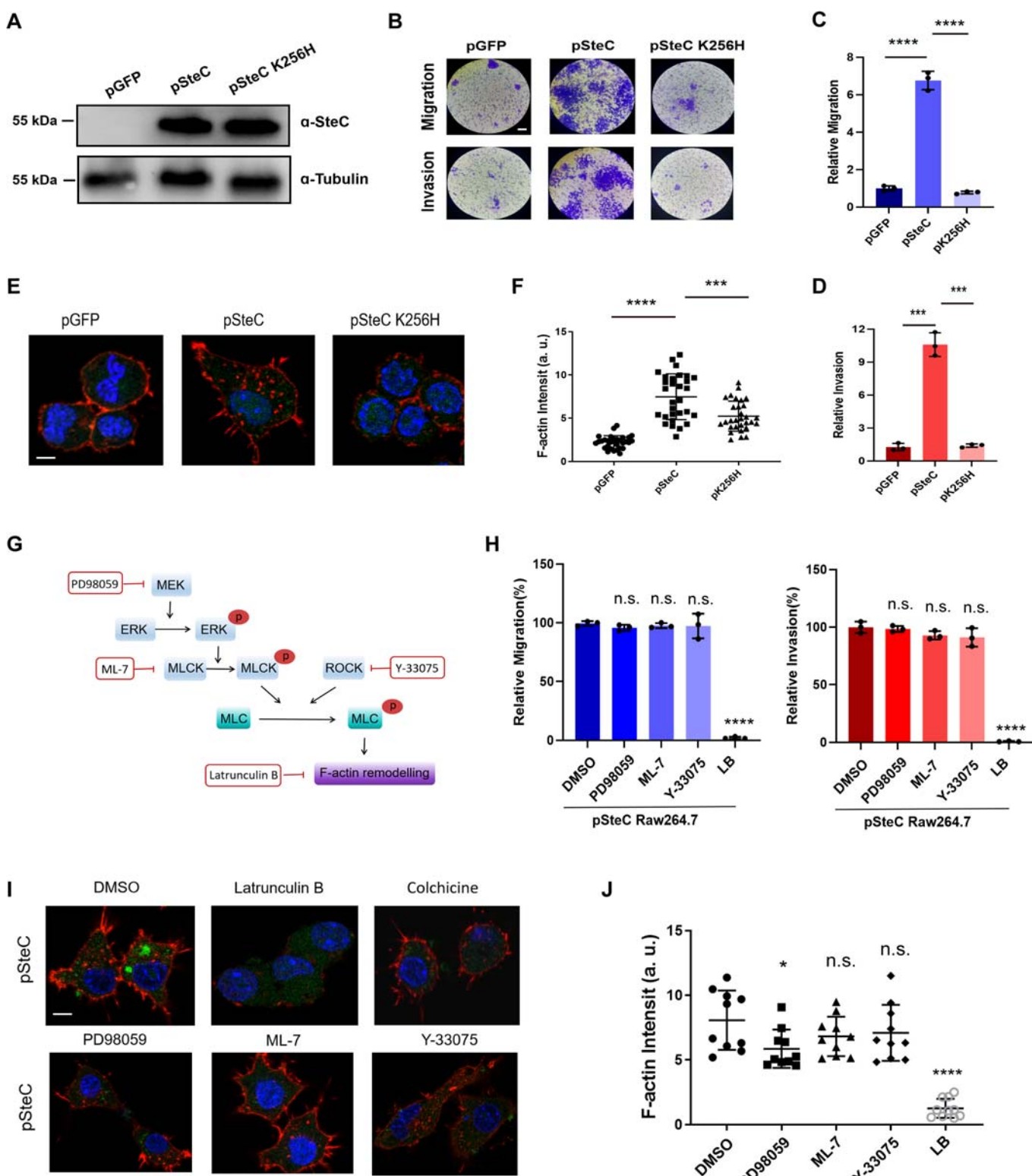

To further explain the specific kinase activity of SteC, we tried to obtain a crystal structure of SteC using the purified fragments. We successfully resolved a structure of SteC 202 aa–375 aa in complex with AMPPCP (Appendix Tab. S2). Unexpectedly, the structure of the SteC kinase domain presents a tightly dimeric state (Fig. 6C). In this structure, AMPPCP molecule undergoes hydrolysis, leaving only AMP in the binding pocket. To further investigate the catalytic mechanisms of SteC, structure-based molecular dynamics simulation was performed for SteC with ATP or UTP and $Mg^{2+}$ (Appendix Fig. S5A–E). The MD results showed that both ATP

◄

**Figure 3.  SteC promotes macrophage migration and invasion and actin rearrangement in a MEK/MLCK-independent manner.**

(A)The expression of SteC were detected in RAW264.7 cells expressing GFP (pGFP), SteC (pSteC), or SteC K256H (pSteC K256H) using western blotting. (B–D) The migration and invasion of pGFP, pSteC, or pSteC K256H. The cells that migrated to the lower chamber were stained and counted under light microscopy. Scale bars, 50 μm. The amounts of migration and invasion of pGFP were set as 1. (E) RAW264.7 cells expressing GFP, SteC, or SteC K256H were stained for F-actin (red) and DAPI (blue). Scale bar, 5 μm. (F) The fluorescence intensity of intracellular F-actin related to (D) was quantified. More than 20 individual cells were measured for each group. (G) Schematic of traditional MLC activation pathway and corresponding inhibitors. PD98059: MEK inhibitor, ML-7: MLCK inhibitor, Y-33075: ROCK inhibitor, Latrunculin B: actin polymerization inhibitor. (H) The migration and invasion of pSteC with or without four inhibitors. The amounts of migration and invasion of pSteC without inhibitors (DMSO) were set as 100%. (I) Four indicated inhibitors were added to pSteC to determine their impact on F-actin rearrangement. F-actin (red), DAPI (blue), GFP (Green). Scale bar, 5 μm. (J) The fluorescence intensity of intracellular F-actin related to (I) was quantified. More than 20 individual cells for each group were measured. Treatments were normalized to DMSO control. Data information: The mean ± SD of three biological replicates was shown. An unpaired *t*-test was used to determine the statistical significance between the two groups. Unless otherwise specified, the infected group was compared with the uninfected control group. *$P < 0.05$, ***$P < 0.001$, ****$P < 0.0001$, n.s. not significant. Source data are available online for this figure.

and UTP well fit in the hydrophobic pocket of of SteC dimer (Appendix Fig. S5F–K). Unlike traditional kinases that bind ATP through the N-lobe and C-lobe from a single molecule (Appendix Fig. S6A,B), the ATP molecule was fixed at the dimeric interface of SteC, where Ile238, Val254, Tyr314 and Ile315 from molecule A are involved in adenine-binding and Asn350, Asp362 and Asp364 from molecule B act as the catalytic center for phosphate ion transfer (Fig. 6C). Asp364 is the only basic residue near the γ-phosphate of ATP, so we hypothesized that Asp364 might act as the general base for nucleophilic attack (Fig. 6D). The structure of SteC suggests that the kinase activity of SteC occurs via dimerization (Fig. 6E). Compared with conventional kinases, fewer hydrogen bonds formed between SteC and the ATP molecule, which may explain why SteC exhibits low NTP selectivity (Appendix Fig. S5H, I). Sequence alignment showed strict conservation of Asn350, Asp362, and Asp364 across species (Fig. 6F). The exact role of these residues was investigated further by mutagenesis (Fig. 6G).

A previous study found that the K256H mutation of SteC eliminates its kinase activity (Odendall et al, 2012). Structural analysis demonstrates that Lys256 is located in the core region of the SteC dimer and interacts with Asp362 through electrostatic interactions (Appendix Fig. S6C) in the active center of the enzyme. According to the structure, the K256H mutation would directly affect interaction with Asp362, blocking ATP binding.

Phylogenetic trees were generated based on alignments of SteC amino acid sequences (Appendix Fig. S7). SteC is prevalent in all *Salmonella* species. SteC is also present in other pathogenic bacteria such as *Saphylococcus, Yokenella, Cedecea, Erwinia*, and *Providencia*. The essential residues in the kinase active center of SteC are strictly conserved across those species, suggesting a uniform mechanism for SteC-mediated phosphorylation.

## The 168 aa–202 aa helix of SteC is crucial for Myl12a recruitment

The interaction between SteC and Myl12a was further investigated by testing the interaction between Myl12a and the eight SteC fragments. The SteC 168 aa–364 aa fragment and Myl12a exhibit similar binding strength to that of full-length SteC. However, SteC 194 aa–364 aa has a tenfold decreased affinity with Myl12a, and no binding was detected for the 202 aa–364 aa fragment (Fig. 6H), indicating that M-helix (168 aa–202 aa) is crucial for SteC–My12a interaction. Structural analysis of myosin (PDB code:7MF3) showed that the binding of myosin light chain to myosin heavy chain also relies on a long α-helix. We speculate that SteC recruits Myl12a by mimicking the substrate of Myl12a using the M-helix.

Molecular dynamics simulation was performed with Myl12a and the predicted structure of SteC 168 aa–375 aa (Appendix Fig. S6D–G; Tab. S3). There are two critical portions of SteC (168 aa–202 aa helix and 336 aa–358 aa regions) that may participate in the interaction with Myl12a (Fig. 6I). We mutagenized residues in these parts of the SteC protein and found the L186A/I187A/D188A and D188A/F189A/L190A mutations lost the ability to interact with Myl12a, confirming our recruitment hypothesis (Fig. 6J). The high degree of conservation of residues in the M helix is specific to *Salmonella* and not observed in other species (Appendix Fig. S7C), implying that the interactions between SteC and Myl12a are unique to *Salmonella*.

## SteC is crucial for breaching the intestinal vascular barrier and dissemination during infection

Different studies have yielded conflicting results on the contribution of SteC to intracellular growth and virulence (Buckner et al, 2011; Geddes et al, 2005; Heggie et al, 2021; Morgan, Campbell et al, 2004). We measured the survival and growth of WT and Δ*steC* strains inside RAW264.7 macrophages by CFU counting method and electron microscopy (Fig. 7A, B). The results demonstrated that the absence of SteC resulted in an enhanced ability of *Salmonella* to proliferate within macrophages, which is in line with a prior report by Odendall (Odendall et al, 2012).

Wild *Salmonella* have been observed to accumulate in CX3CR1⁺ macrophages that surround the microvessels of the small intestine (Fig. 1A–C). Then, the location of Δ*steC* strains of early infection was determined by immunofluorescence. Interestingly, we observed that the Δ*steC* strains were also enriched in macrophages around the blood vessels 1 h post-infection, accounting for ~30% of the bacteria present in these macrophages (Fig. 7C). However, the quantitative results of bacteria in the blood and liver showed that Δ*steC* strains displayed a temporary inability to infiltrate the bloodstream and tissues 1 h post-infection (Fig. 7D,E). Spadoni et al, have described the existence of a gut-vascular barrier (GVB) in the host intestine that prevents bacteria from migrating from the gut into blood circulation (Spadoni et al, 2015). Because plasmalemma vesicle-associated protein-1 (PV-1) acts as a marker of impaired GVB (Bertocchi et al, 2021), so the PV-1 level in the small intestine of WT or Δ*steC*-infected mice was detected by immunofluorescence staining (Fig. 7F). Although the expression of PV-1 significantly increased after *Salmonella* infection, no significant difference was observed in the expression level of PV-1 between WT and Δ*steC* groups, suggesting the difference in tissue dissemination between WT and Δ*steC* is not caused by GVB

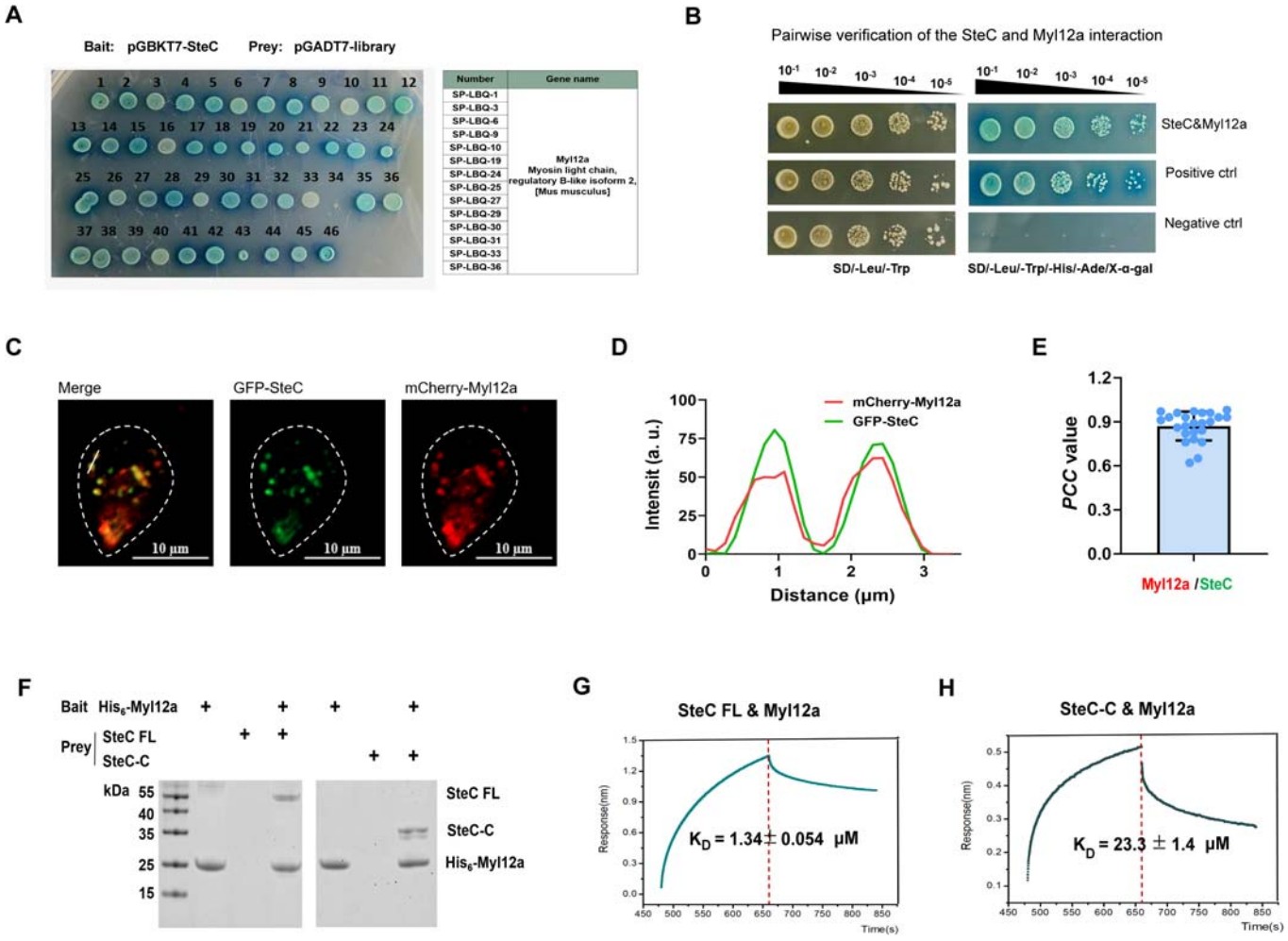

**Figure 4. SteC directly interacts with Myl12a in vivo and in vitro.**

(A) Yeast two-hybrid screening of SteC-interacting proteins of macrophage. SteC was used as the bait to screen a RAW264.7 AD library. The sequencing results showed that fourteen positive clones corresponded to Myl12a. (B) Pairwise verification of the SteC-Myl12a interaction. (C) Representative confocal images show GFP-SteC co-localizes with mCherry-Myl12a. Scale bar, 10 μm. (D) The fluorescence signal intensity on the white dashed line was plotted. (E) Pearson's colocalization coefficient (PCC) value of SteC-Myl12a colocalization from 20 different cells. The mean ± SD of 20 individual cells was shown. (F) Pull-down assays of His₆-Myl12a and SteC FL and SteC-C were performed. SteC FL: full-length SteC; SteC-C: SteC 194 aa–457 aa. (G, H) The binding affinities of SteC FL or SteC-C and Myl12a were measured by BLI. Source data are available online for this figure.

damage. Given that SteC enhances the motility and invasion ability of macrophages, we speculated that SteC might be associated with the ability of *Salmonella* to breach GVB. To test this hypothesis, WT *Salmonella* and Δ*steC* strains expressing luciferase were constructed, and animal infection experiments were performed using oral inoculation with nude mice. Fluorescence imaging was performed of the infected mice in a real-time fashion. The results showed that WT *Salmonella* spread to the intestine and other organs earlier than Δ*steC* (Fig. 7G). More importantly, WT-infected mice showed multi-organ metastases of bacteria, while the infection of Δ*steC*-infected mice was mainly concentrated in the small intestine, indicating that SteC is involved in breaking through the vascular barrier and spreading of *Salmonella* (Fig. 7G). Pathological analysis of infection results also revealed that 20 h post-oral gavage infection, Δ*steC* were frequently detected in cells surrounding blood vessels in intestinal tissue, whereas such occurrences were barely

observed in wild-type cases (Fig. 7H,I), supporting our hypothesis that WT *Salmonella* can infiltrate the bloodstream through macrophages, while Δ*steC* fail to trigger macrophages to enter the bloodstream and therefore remain around intestinal blood vessels.

In order to further verify the ability of infected macrophages to penetrate blood vessels, we conducted an in vitro experiment simulating vascular penetration. We immobilized vascular endothelial cells in a chamber and observed the penetration of these cells by macrophages (Fig. 7J). The results indicated that uninfected macrophages had no ability to penetrate the blood vessels. However, macrophages infected with wild-type *Salmonella* exhibited a strong ability to traverse the vascular endothelial cells, while the ability of Δ*steC* knockout bacteria to penetrate the vascular endothelial cells was significantly reduced compared to the wild-type (Fig. 7K). Furthermore, experiments using stable SteC-expressing cells also demonstrated that cells expressing SteC rapidly

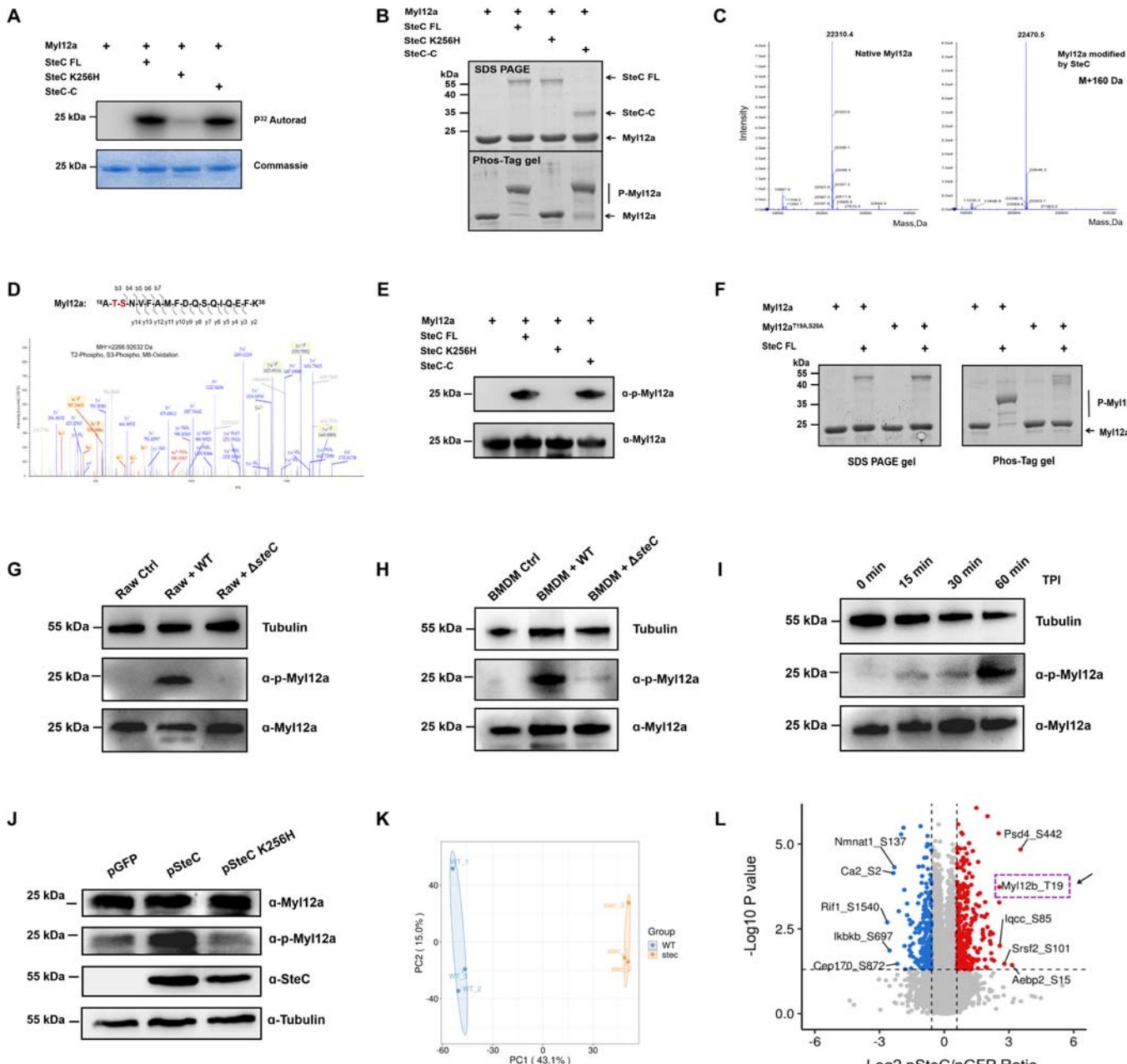

**Figure 5. SteC phosphorylates and activates Myl12a in vivo and in vitro.**

(A) In vitro phosphorylation assays of Myl12a catalyzed by SteC using γ$^{32}$P-ATP. The experiments were performed using purified Myl12a and SteC full-length protein or protein fragments. $^{32}$P Autorad, autoradiography of $^{32}$P-labeled proteins. (B) In vitro phosphorylation assays of Myl12a catalyzed by SteC using Phos-tag gel. P-Myl12a: Phosphorylated Myl12a. (C) Mass spectra of Myl12a and Myl12a phosphorylated by SteC confirm that the molecular weight of SteC-modified Myl12a increased by 160 Da compared to the native Myl12a. (D) MS/MS spectra of phosphorylated peptides of SteC-modified Myl12a. The b and y ions are indicated along the peptide sequence above the spectra. (E) Reaction products of in vitro phosphorylation of Myl12a were immunoblotted with the indicated antibodies. (F) In vitro phosphorylation assays of Myl12a T19AS20A by SteC using Phos-tag gel. P-Myl12a: Phosphorylated Myl12a. (G) The phosphorylation levels of Myl12a during infection were detected using western blotting in RAW264.7 cells 1 h post-WT or Δ*steC* infection. (H) The phosphorylation levels of Myl12a during infection were assessed using western blotting in BMDM cells 1 h post-WT or Δ*steC* infection. (I) The phosphorylation levels of Myl12a in BMDM was quantified using western blotting at different times after infection with wild-type *Salmonella*. (J) The phosphorylation level of Myl12a in the indicated cells was quantified using western blotting. (K) The phosphoproteomic data determined by LC-MS/MS was assessed by principle-component analysis (PCA). WT: pGFP, STEC: pSteC. (L) Volcano plot for differentially phosphorylated peptides identified in pSteC and pGFP. Student's *t*-test was used to determine the statistical significance between the two groups. The top ten differential proteins in phosphorylation are labeled and Myl12b is highlighted. Source data are available online for this figure.

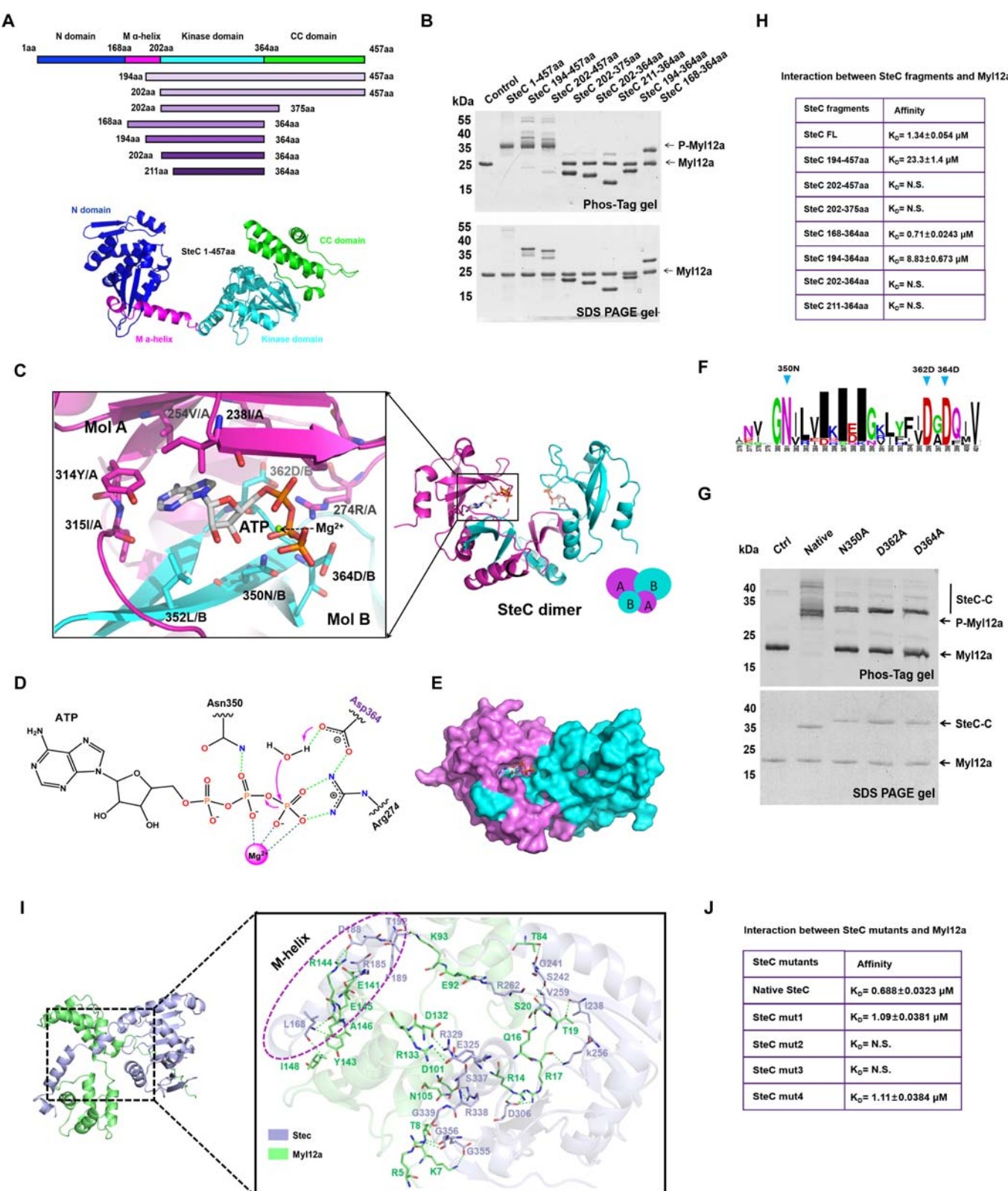

traversed the vascular endothelial cells (Fig. 7L; Movie EV3). The aforementioned findings provide robust evidence to support the pivotal role of SteC in facilitating macrophage infiltration through the vascular barrier.

## Discussion

*Salmonella* utilizes flagella to move to a favorable environment in vitro, but flagella synthesis is shut down when entering host cells

**Figure 6.   Structural insights into the catalytic and Myl12a-recruited mechanisms of SteC.**

(A) The upper panel depicts a schematic representation of SteC's domains and fragments. The lower panel depicts the overall structure of SteC predicted by trRosetta server. (B) In vitro phosphorylation assays were performed with SteC full-length and fragments. The samples were analyzed using Phos-tag gel. P-Myl12a: Phosphorylated Myl12a. (C) The structure of SteC-ATP was obtained from molecular dynamic simulation based on SteC-AMP structure (PDBid: 8JBI). The zoomed view of the SteC-ATP interface is marked with a black frame. The essential residues and ATP were shown in stick mode. (D) The proposed mechanism of SteC-mediated phosphorylation. Asp364 acts as the general base for nucleophilic attack. (E) The dimeric structure of SteC is shown in surface mode. ATP molecule bound in the hydrophobic pocket of SteC dimer. (F) Sequence Logo of the multiple sequence alignment in the active sites of SteC. (G) Activity of SteC or active site mutants to phosphate Myl12a. (H) The binding affinities of SteC fragments and Myl12a were measured by BLI. (I) Model of SteC-Myl12a complex. The zoomed view of the SteC-Myl12a interface is marked with a black frame. The residues from the SteC M-helix of the interface are highlighted by a purple circle. (J) The binding affinities of SteC mutants and Myl12a were measured by BLI. Source data are available online for this figure.

so that *Salmonella* can escape from the host immune system (Li et al, 2017; Stewart et al, 2011). Intriguingly, even in the absence of flagella, *Salmonella* can rapidly infiltrate the bloodstream and propagate to target organs following its invasion of the intestine (Vazquez-Torres et al, 1999; Worley et al, 2006). The mechanisms underlying this phenomenon remain largely unclear.

Macrophages are the main host cells of *Salmonella*. Tissue-resident macrophages possess limited migration capabilities, and many studies have shown that antigen-activated macrophages almost completely lose their motility (Hind et al, 2016; Perez-Rodriguez et al, 2022). However, both previous studies and this study have shown that the migration ability of macrophages is significantly increased after *Salmonella* infection, rather than being inhibited (Worley et al, 2006). This study offers a coherent explanation for the observed phenomenon (Fig. 7M). After entering the host cell, *Salmonella* secretes effector SteC into the host cytoplasm through T3SS-2 and directly phosphorylates and activates MLC in a manner independent of the MEK/ERK/MLCK pathway, promoting the migration and invasiveness properties of macrophages and helping parasitized macrophages break through the vascular barrier and spread to target organs. *Salmonella* can reverse the weakened mobility phenotype of macrophages following infection via the kinase activity of SteC, which leads to a unique cellular phenotype called "enhanced mobility parasitized macrophages." Recently, the *Mycobacterium tuberculosis* effector EsxM was found to promote the migration of macrophages and promote bone metastasis (Saelens et al, 2022). Toxoplasma was reported to promote its dissemination by regulating monocyte tissue migration by ROP17 kinase via the Rho-ROCK pathway and inducing dendritic cell-like migratory properties in infected macrophages through its effector GRA28 (Drewry et al, 2019; Ten Hoeve et al, 2022). The above findings and our work indicate that manipulating the migration properties of macrophages may be a common survival strategy of pathogens.

*Salmonella* SteC was previously reported to regulate actin rearrangement in fibroblasts by activating ERK phosphorylation (Odendall et al, 2012). This mechanism might not occur in macrophages, because the MEK-ERK-MLCK pathway is largely suppressed in macrophages during *Salmonella* infection through the NLRP6 and NLRP12 pathways. The phosphorylation level of ERK in macrophages significantly decreases during *Salmonella* infection (Anand et al, 2012; Zaki et al, 2014). Therefore, SteC can not regulate actin rearrangement through directly phosphorylating ERK. This hypothesis was supported by the finding that treatment with the corresponding inhibitors has almost no effect on the phenotype of SteC-induced migration enhancement of macrophages. Based on our findings, we suggest that the SteC-Myl12a

pathway is the primary mechanism that *Salmonella* employs to regulate macrophage motility. Previously, another SPI-2 effector SseI was found to promote the motility of macrophages and accelerate systemic transmission by interacting with the host protein TRIP6. However, previous studies and our data revealed that SseI knockout *Salmonella* still partially promoted macrophage motility, while SPI-2 deficient bacteria completely lost their ability to drive macrophage motility (Worley et al, 2006), indicating that additional unknown factors in SPI-2 are involved in this regulatory process in addition to SseI. The results of this study identify SteC protein as the other key player in this regulation. SseI can inhibit the migration of dendritic cells to the spleen in vivo by binding to IQGAP1, indicating that SseI may activate or inhibit host cell motility under special conditions (McLaughlin et al, 2009). Two SPI-2 effectors regulate the actin cytoskeleton and motility of macrophages using entirely distinct mechanisms, indicating the importance of host cytoskeleton regulation during *Salmonella* infection. However, additional research is necessary to clarify the time frame for gene expression and functional effects of SteC and SseI.

This work reveals that SteC kinase has unique catalytic characteristics compared to known kinases. SteC can employ all four nucleotides (ATP, CTP, UTP, and GTP) for phosphorylation, while other kinases generally use ATP as a donor. Given that ATP is usually depleted during the anti-inflammatory process (Rostami et al, 2020), the unique enzyme activity of SteC provides *Salmonella* with a direct advantage in ATP competition. Furthermore, the crystal structure reveals a dimerization-mediated catalytic mechanism of SteC, different from all known kinases binding ATP using the N-lobe and C-lobe from a single molecule. Fewer hydrogen bonds were identified between SteC and the adenine ring of ATP, which could explain why SteC does not display NTP selectivity.

Despite these advances in our understanding, there are still some unresolved questions. Our animal infection experiments demonstrated that the Δ*steC* strains exhibit higher toxicity than the wild strains (unpublished data). In addition to its cytoskeleton-related function, Myl12a acts in immunomodulation and phagocytosis, and SteC may also activate Myl12a's other functions (Barger et al, 2022; Hayashizaki et al, 2016; Stendahl et al, 1980). More importantly, in addition to My112a, other host substrates were identified during the yeast two-hybrid and phosphoproteomic screening, indicating that SteC may also work in a way that is independent of Myl12a. Additionally, SteC itself acts as an antigen for host cells, which may directly trigger the host immune system. Therefore, SteC may be a versatile protein with the ability to impact various host functions by regulating different targets. Additional

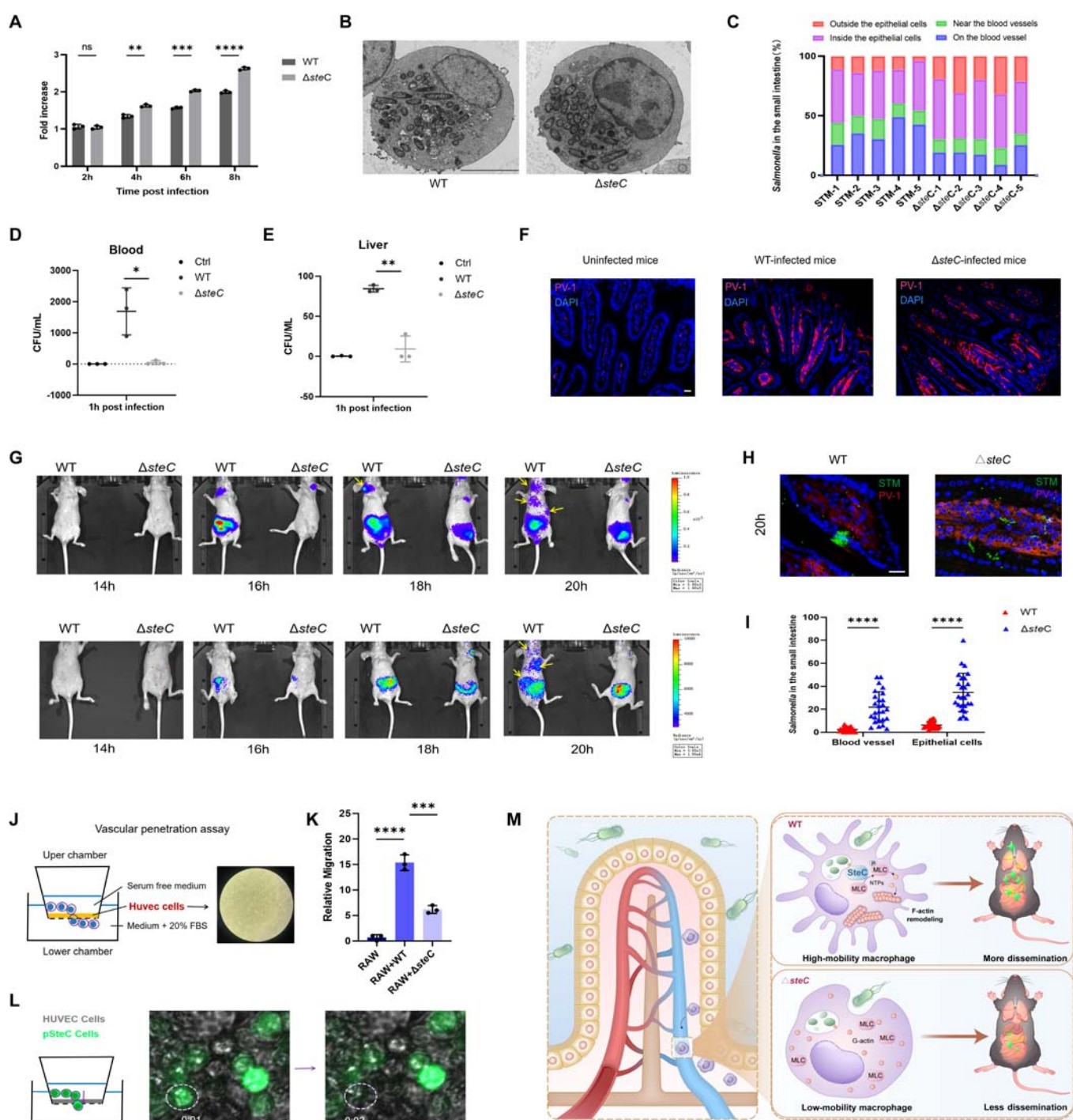

investigation is required to uncover the other potential regulatory functions of SteC.

Overall, the results of this study revealed a special kinase secreted by intracellular bacteria that directly activates MLC and enhances macrophage motility. This activation process does not rely on the traditional cascade reaction process of MEK-ERK-MLCK, and can be achieved using nucleotides other than ATP as substrates, which is a novel way to bypass competition with host cells. Animal studies have demonstrated that the deletion of SteC in *Salmonella* significantly impairs their ability to spread throughout host organs. Therefore,

developing drugs that target SteC could potentially offer new treatment options for individuals with typhoid fever.

## Methods

### Cell lines and culture conditions

The RAW264.7 macrophage cell line, RAW264.7 expressing SteC (pSteC), RAW264.7 expressing SteC K256H (pSteC K256H),

**Figure 7. SteC is indispensable for gut-vascular barrier breakthroughs and systemic spreading during *Salmonella* infection.**

(A) The proliferation of WT or Δ*steC* strains in RAW264.7 cells were measured by counting CFU. The bacterial count after a two-hour invasion was set to 1 for the WT or Δ*steC* group, respectively. $n = 3$ per group. (B) Representative images of intracellular bacteria by electron microscopy. RAW264.7 cells were observed 10 h post-infection with WT or Δ*steC* strains. Scale bar, 5 μm. (C–E) C57BL/6 mice were orally infected with WT or Δ*steC* for 1 h. (C) Statistics of *Salmonella* distribution in host small intestinal after infection. $n = 5$ per group. (D, E) The number of indicated bacteria in the blood (D) or liver (E) 1 h post-infection. $n = 3$ per group. (F) The small intestinal of infected mice were analyzed through immunofluorescence. BALB/C mice were orally infected with WT or Δ*steC*. Small intestinal (SI) blood vessels were stained with PV-1 (Red). DAPI (blue). *Salmonella* (Green). Scale bars, 10 μm. (G) In vivo bioluminescence imaging of BALB/C-nu mice inoculated intragastrically with WT or Δ*steC* expressing luciferase. Representative luminescence images show WT *Salmonella* spreads to the liver, spleen, and lymph nodes earlier than Δ*steC*. (H) Representative images depicting the distribution of WT or Δ*steC* in the small intestine after a 20-hour infection. PV-1 (Red). DAPI (blue). *Salmonella* (Green). Scale bars, 10 μm. (I) Quantification and distribution of *Salmonella* within small intestinal cells at 20 h post-infection in each experimental group analyzed by 25–30 visual fields of three mice. (J) Schematic diagram illustrating the vascular penetration assay. (K) HUVEC cells were seeded into the transwell chamber and allowed to incubate overnight, followed by fixation with 4% paraformaldehyde. RAW264.7 cells infected by WT or Δ*steC* were added and co-cultivated for 24 h. The cells that migrated to the lower chamber were stained and counted under light microscopy. $n = 3$ per group. The uninfected RAW264.7 cells were used as negative controls. (L) Representative imaging results SteC-expressing RAW264.7 cells penetrating the HUVEC cell barrier. Scale bars, 10 μm. (M) Schematic model of SteC-mediated *Salmonella* dissemination. Data information: The mean ± SD was shown. An unpaired *t*-test was used to determine the statistical significance between the two groups. Unless otherwise specified, the infected group was compared with the uninfected control group. *$P < 0.05$, **$P < 0.01$, ***$P < 0.001$, ****$P < 0.0001$, n.s. not significant. Source data are available online for this figure.

RAW264.7 expressing GFP (pGFP), Myl12a knockout RAW264.7 (Myl12a$^{-/-}$) and HeLa cells were cultured in Dulbecco modified Eagle medium (DMEM, Gibco) containing 10% fetal bovine serum (FBS) at 37 °C and 5%(v/v) $CO_2$. The RAW264.7 overexpressing cell lines (pSteC, pSteC K256H, pGFP) were constructed by Ubigene (Guangzhou, China).

Bone marrow cells were harvested from the femurs and tibias of C57BL/6 mice aged 6–8 weeks. The cells were then cultured in a complete DMEM medium containing 20 ng/mL GM-CSF and 10% FBS at 37 °C with 5%(v/v) $CO_2$. On the third and sixth day of culture, an additional 2–3 mL of complete medium was added. On the seventh day, if the cells are fully adherent, subsequent experiments can be conducted.

## Bacterial strains and propagation

*S. typhimurium* ATCC14028 culture was used for functional studies. The culture was grown overnight and subcultured into an LB medium with corresponding antibiotics. When the OD600 reached 0.4–0.5, bacterial cells were harvested for subsequent functional study. Proteins were expressed in *E. coli* BL21(DE3). The culture was grown overnight and subcultured into an LB medium with corresponding antibiotics. When the OD600 reached 0.4–0.6, cultures were cooled to 16 °C and induced overnight by IPTG addition.

## Macrophage infection models

Macrophage (RAW264.7 or BMDM) cells were cultured in a 10% fetal bovine serum (DMEM) medium. About $10^7$ cells were inoculated into a 100 mm cell culture dish. Cells were infected with logarithmic wild-type *Salmonella* and Δ*steC* at the indicated MOI. After 1 h of infection, the cells were washed twice with PBS, and 100 μg/mL of fresh DMEM medium containing gentamicin was added to kill extracellular bacteria. After 2 h of infection, the cells were replaced with DMEM medium containing 10 μg/mL gentamicin and continued to be cultured in a 37 °C incubator containing 5% (v/v) $CO_2$. Samples were used for further experiments. At specific time points after infection, appropriate cell samples were taken and subjected to gradient dilution before being plated for quantitative analysis (CFU). Assays of the

migration and invasion of the infected cells are outlined in the method details section.

## Murine infection models

All animal experiments were approved by the Animal Care and Use Ethics Committee of the Institute of Basic Medicine, Shandong Academy of Medical Sciences (IBMSAMSC Number 087). In the experiments of Figs. 1 and 7C, female C57BL/6 mice aged 7–9 weeks were fasted for 3 h before orogastric infection with $2 \times 10^8$ CFU of GFP labeled *S. typhimurium* in 200 μL of PBS. One hour after infection, peripheral blood was collected in heparinized tubes. For animal imaging (Fig. 7G), female nude mice (BALB/c-nu) aged 4 weeks were selected for animal experiments. Mice in each group ($n = 3$) were orally infected with $1 \times 10^6$ CFU of indicated *Salmonella* expressing luciferase. D-Luciferin potassium salt was intraperitoneally injected into mice at the indicated time point post-infection. Then, bioluminescence imaging was performed using IVIS Spectrum CT. For the data shown in Fig. 7F–H, 14–16 g female BALB/c mice were selected, with each group of mice ($n = 3$) orally infected with $2 \times 10^8$ *Salmonella*. At the indicated time points after infection, peripheral blood was obtained. Liver tissue was lysed with physiological saline containing 1% Triton X-100 (PBST). All samples were diluted with PBST, applied to an LB solid plate, and cultured at 37 °C for 48 h.

## Construction of *Salmonella* knock-out strains

*S. typhimurium* ATCC14028 strain was purchased from the American Type Culture Collection. The knockout strains (Δ*steC*, Δ*ssaW*, and Δ*sseI*) were constructed using the lambda Red recombinase system as described previously (Datsenko and Wanner, 2000). Briefly, the chloramphenicol resistance-FRT cassette was amplified from pKD3 with sequences that flank the corresponding gene. The amplified product was transformed into *S. typhimurium* ATCC14028 strain containing the pKD46 plasmid, and then recombined colonies were selected on LB agar plates containing 25 μg/ml chloromycetin. PCR was performed with two primers outside the gene of interest to confirm the deletion of the corresponding gene. The pCP20 plasmid was used to remove the chloramphenicol resistance gene.

## Plasmid construction

Primers and plasmids used in this study are listed in Appendix Tables S4, S5. For the in vivo study, SteC and SteC K256H were separately cloned into the YOE-LV001 vector (Ubigene). For the Y2H study, SteC and Myl12a were cloned into pGBKT7 and pGADT7 vectors, respectively (OE Biotech). For colocalization experiments, SteC and Myl12a were cloned into pEGFP-C1 and pmCherry-C1 vectors, respectively. For biochemical study, genes encoding SteC full-length protein, or SteC 194–457 aa, 202–457 aa, 202–375 aa, 202–364 aa, 211–364 aa, 194–364 aa, 168–364 aa fragments were amplified from *S. typhimurium* genomic DNA and cloned into an expression vector, pGL01. Genes of murine Myl12a, Myl12b, and Myl9 were synthesized and cloned into a pGL01 vector. Mutants of SteC (K256H, N350A, D362A, D364A, I178A/ Y179A/E180A, F182A/E183A/Q184A/R185A, L186A/I187A/ D188A, and D188A/F189A/L190A) and Myl12a T19A,S20A were separately cloned into pGL01 using Mut Express II Fast Mutagenesis Kit V2 (Vazyme).

## Cell migration and invasion assays

Cell migration assays were performed using transwell chambers with 8 μm polycarbonate filters. Cells were seeded in 24-well plates at a density of $2.5 \times 10^5$ cells/well. The cells were seeded in a serum-free medium in the upper transwell chamber, and 20% fetal bovine serum was added to the lower chamber. For the experiments in Fig. 2, RAW264.7 cells were infected with logarithmic wild-type *Salmonella* or knockout strains at the indicated MOI. After 1 h of infection, the cells were washed twice with PBS and 100 μg/mL of fresh DMEM medium containing gentamicin was added to kill extracellular bacteria. After 2 h of infection, the cells were replaced with DMEM medium containing 10 μg/mL gentamicin and continued to be cultured in a 37 °C incubator containing 5% (v/v) $CO_2$. After 8 h of culture, cells were suspended in serum-free DMEM medium and 200 μL of cell suspension was added to each well in the upper chamber of a 24-well plate. After 24 h of incubation at 37 °C, cells that had migrated to the lower surface were fixed by 1% crystal violet, counted, and photographed (400×). Detection of cell invasion via matrigel-coated transwell assays. The experiments were performed using transwell chambers with 8 μm polycarbonate filters with 60 μl diluted matrix gum (ABW) added to the upper transwell chamber.

For the experiments in Figs. 2A,C and 3H, after seeding the cells in the upper transwell chamber, chemokines or inhibitors were added at final concentrations of 10 ng/mL for CCL2, 100 ng/mL for CCL7, 10 μM for PD98059, ML-7, Y-33075, Latrunculin B, and Colchicine. After 24 h of incubation at 37 °C, cells that had migrated to the lower surface were fixed by 1% crystal violet, counted, and photographed (400×).

## Motion-track analysis

Approximately $5 \times 10^5$ BMDM cells were seeded in a 35 mm dish, infected at an MOI of 20:1, and washed with PBS 1 h after infection to remove extracellular bacteria, then replaced with DMEM medium containing 100 ug/mL gentamicin. One hour after infection, time-lapse photography was performed using a Cell

Discoverer 7 (Zeiss) laser confocal microscope, taking one frame every 10 s. The "Track Spots" function of Imaris software was used for analysis, and the diameter of the target to be tracked was set according to the cell size to obtain the cell motion track and the coordinates at each moment. The data were then processed in Excel by subtracting the coordinates of each cell from the coordinates of the corresponding cell at the first time point. The processed data were then plotted using GraphPad.

## Co-transfection experiments

HeLa cells were cultured in DMEM containing 10% fetal bovine serum (FBS) at 37 °C and 5% $CO_2$. About $5 \times 10^6$ HeLa cells were inoculated in each well of the six-well plate. When the cell convergence reached 70–90%, Lipfecamine 3000 reagent (Invitrogen) and corresponding plasmids (2.5 μg) were added to the cells according to the manufacturer's instructions. Colocalization experiment was performed 24 h post transfection.

## Fluorescence microscopy

For the experiments in Fig. S1A, the *Salmonella*-infected RAW264.7 cells were washed with a pre-warmed medium. Confocal time-lapse sequences were captured every 20 min interval in real time at 5% (v/v) $CO_2$ level and 37 °C. For the experiments in Figs. 2G and 3E,I, RAW264.7 cells infected with *Salmonella* or treated with inhibitors were fixated with 4% paraformaldehyde for 20 min. After permeabilization with 0.5% Triton x-100 for 15 min, the cells were incubated for 45 min with Actin-tracker Red (Beyotime). Finally, the cells were incubated with DAPI for 10 min. For the experiments in Fig. 4C, the real time localization of SteC and Myl12a was performed using green fluorescent protein (GFP) and mCherry, respectively, fused to protein-expressing plasmids. Green and red channels were recorded to determine the intracellular localization of SteC and Myl12a. These samples were imaged using a Zeiss Celldiscoverer7 or Zeiss LSM 980 confocal microscope and processed with Zeiss ZEN Blue or ImageJ softwares.

## F-Actin quantification

For quantitative analysis of F-actin, the fluorescence images were processed using ImageJ software. Each cell in the image was first extracted and outlined to obtain its cell area. The normalized F-actin intensity was calculated by measuring the fluorescence intensity in each cell and dividing it by the cell area. Twenty cells from each group were collected for analysis, and the difference in F-actin between each group was counted.

## Flow cytometry

Peripheral blood was isolated with a mouse peripheral blood lymphocyte (PBMC) isolation kit (TBDscience). PBMC cells were incubated with Fc block, Zombie NIR to identify live leukocytes and fluorescence-labeled Abs against F4/80, CD45, CX3CR1, and CD11c. Cell surface fluorescence was measured on a FACSymphony flow cytometer (BD Biosciences), and the data were analyzed using Flowjo software (Tree Star).

## Immunofluorescence assay

For the experiments in Fig. 1L, 1 h post-infection, mice were anesthetized with pentobarbital sodium, and peripheral blood samples were collected from their eyeballs. The blood samples were subjected to red blood cell lysis (Solarbio), washed twice with PBS, and then resuspended in DMEM containing 10% FBS. The cells were added onto climbing slides placed in dishes and allowed to adhere for 3 h, then fixed with 4% paraformaldehyde overnight. After permeabilization with 0.2% Triton X-100 for 5 min, the cells were treated with 5% BSA buffer at 37 °C for 1 h. Then the cells were incubated with primary antibodies against *Salmonella* (1:4000) and F4/80 (1:4000) at 4 °C overnight, and incubated with secondary antibodies, Alexa Fluor 594 conjugated Goat anti-rabbit (1:200) and Alexa Fluor 488 conjugated goat anti-rat (1:4000), at room temperature for 1 h. Finally, the slides were incubated with hoechst (1:1000) for 5 min. The cells were imaged using a Zeiss LSM 980 confocal microscope and processed with Zeiss Zen Blue software. Antibody information: *Salmonella* (Abcam ab35156), F4/80 (Abcam ab6640).

For the experiments in Figs. 1A–C, 7F,H, frozen small intestine sections were washed with PBS and blocked in 3% BSA/PBS for 1 h at room temperature. Then the sections were incubated with an anti-PV-1 antibody (1:2000, Servicebio GB113141) or anti-F4/80 antibody (1:500, CST 70076 s) overnight at 4 °C. After washing, the sections were incubated with Cy3-conjugated goat anti-rabbit IgG. For the experiments in Figs. 1A–C and 7H, the sections were heated in antigen repair solution and then incubated with anti-*Salmonella* antibody (1:500, Abcam ab35156) overnight at 4 °C. After washing, sections were incubated with FITC-conjugated goat anti-mouse IgG at room temperature for 10 min. Nuclei were stained with DAPI. Fluorescence images were obtained as indicated above.

## Statistical analysis of the distribution of *Salmonella*

Ten slice results of each mouse were captured under 63x fields using CaseViewer software, and the RGB Stack function of ImageJ was used to separate the channels. After adjusting the contrast and threshold of the *Salmonella* channel, analysis and counting were performed.

## Yeast two-hybrid assay

To identify the SteC-interacting proteins from RAW264.7, a Gateway AD prey library of RAW264.7 cells was first constructed (OE Biotech Company, Shanghai, China). Briefly, the total RNAs of RAW264.7 cells were extracted using TRIzol (Invitrogen), and cDNA was synthesized using the CloneMiner II cDNA Library Construction Kit (Invitrogen) according to the manufacturer's instructions. The cDNAs were recombined with the pDONR222 vector and transformed into *Escherichia coli* DH10B to generate the cDNA library. For screening, the pGBKT7-SteC bait vector and the pGADT7 prey plasmids of the AD library were co-transformed into Y2HGold yeast cells and screened on SD/-Leu/-Trp/-His/X-α-gal plates (DDO/X/A). Blue clones were then grown on SD/-Leu/-Trp/-His/-Ade/X-α-gal/AbA plates (QDO/X/A). The positive clones were isolated and sequenced. To detect the interaction between SteC and Myl12a, SteC and Myl12a were cloned into pGBKT7 and pGADT7, respectively. The pGBKT7-SteC and pGADT7-Myl12a plasmids were co-transformed into yeast strain Y2HGold cells, and cultured on DDO/X/A and then QDO/X/A plates. Positive and negative controls were included.

## Recombinant expression and purification

Proteins were expressed in *Escherichia coli* BL21 (DE3) as described above. Cells were collected and lysed through ultrasonic treatment. Target proteins were purified using Nickel-NTA affinity columns, and the His-tag was removed through PPase treatment (Li et al, 2017). Next, the proteins were concentrated and purified utilizing Superdex 200 chromatography. Selenomethionine-labeled SteC was expressed in *E. coli* BL21 (DE3) using methionine biosynthesis inhibition methods (Hendrickson et al, 1990).

## Pull-down assay

Purified Myl12a with N-terminal 6×his-tag was used as a bait protein. Purified SteC FL or SteC-C protein (at 5-fold the concentration of Myl12a) was passed over columns with or without Myl12a immobilized onto the Nickel (Ni)-NTA resin. The Ni$^{2+}$ affinity chromatography column was thoroughly washed to remove unbound proteins using Wash Buffer (25 mM Tris-HCl pH 8.0, 100 mm NaCl), and then eluted using Elution Buffer (25 mM Tris-HCl pH 8.0, 100 mM NaCl, 200 mM imidazole) and the eluate was analyzed by SDS-PAGE.

## Biolayer interferometry assays (BLI)

Biospheric interferometry experiments were performed at 25 °C using the Octet RED96 instrument. Purified Myl12a was incubated with EZ-Link biotin (MCR = 1:1) at 28 °C for 30 min, and then desalted on a Mini Trap G-25 column (Cytiva). After the baseline step, the streptavidin sensor (Sartorius) was saturated with Myl12a for 300 s. After PBS equilibration, the sensor was loaded into a buffer containing the indicated concentration of SteC/SteC fragments or mutants (0.1–10 μM) to measure the binding signal, and then the sensor was loaded into PBS buffer to detect dissociation. SteC was biotinylated and used as the sensor ligand, and Myl12a, Myl12b, and Myl9 were tested as analytes.

## Western blotting

Western blotting was performed using standard methods. In brief, whole cell extracts were prepared using Beyotime cell lysate buffer (containing protease inhibitor and phosphatase inhibitor). The samples were analyzed on 12% NuPAGE gels (Thermo Fisher). After electrophoresis, proteins were electroeluted at 120 V onto a polyvinylidenedifluoride (PVDF) membrane (Pall). The PVDF membrane was first blocked in 5% skim milk powder and then incubated with the indicated primary antibody overnight at 4 °C, followed by 1 h with a secondary antibody at room temperature. Protein bands were visualized by an enhanced chemiluminescence assay kit (Millipore). Antibody information: SteC (Dia-An BiotechCustom made), Tubulin (Abcam ab176560), Myl12a (Proteintech 16287-1-AP), Phosphorylated Myl12a (Abcam ab126739).

## In vitro phosphorylation experiments with γ³²P-ATP

For the data shown in Fig. 5A, 20 μg of purified Myl12a and 5 μg of SteC were incubated in 100 μl reaction buffer containing 25 mM Tris-HCl (pH 7.5), 1 mM DTT, 10 mM MgCl₂/MnCl₂ and 2 μCi gamma-32p-ATP (Perkin Elmer) at 30 °C for 30 min. Next, 10 μl of the product was analyzed using a 12% NuPAGE gel. The radiolabeled bands were visualized by autoradiography and then the gels were stained using Coomassie bright blue.

## Analysis of in vitro phosphorylation experiments by Phos-tag gels

For the data shown in Fig. 5B,E and Figs. S2, S4, 4 μg of purified Myl12a/Myl12b/Myl9/Myl12a mutants and 1 μg of SteC/SteC fragments were incubated in 20 μl reaction buffer containing 25 mM Tris-HCl (pH 7.5), 1 mM DTT, 10 mM MgCl₂/MnCl₂, and 100 μM ATP/UTP/CTP/GTP at 30 °C for 30 min. The products were then analyzed using Phos-tag gels (100 μM phos-tag chemical compound in 12.5% acrylamide gel) and the gels were stained using Coomassie bright blue.

## Kinetic analysis of SteC kinase using NTP

Purified Myl12a (20 μg) and SteC FL (5 μg) were incubated in a 100 μl reaction buffer containing 25 mM Tris-HCl (pH 7.5), 1 mM DTT, 10 mM MnCl₂ and the indicated concentration of NTP at 30 °C for 30 min. Next, 10 μl of the product was analyzed using Phos-tag gels (100 μM phos-tag chemical compound in 12.5% acrylamide gel). The assays were repeated three times for each group. The consumption of Myl12a at the indicated reaction was analyzed by measuring the intensity of signal corresponding to unmodified Myl12a by ImageJ. Data were statistically analyzed using GraphPad, and $K_m$ and $V_{max}$ values were calculated using Michaelis–Menten curve fitting.

## Sample preparation for phosphoproteomics

The pSteC and pGFP RAW264.7 cells were collected and disrupted by ultrasonication on ice and then digested with trypsin overnight. The samples were dissolved in a buffer containing 50% acetonitrile and 0.5% acetic acid and then incubated with IMAC (Immobilized Metal-Affinity Chromatography). After thorough washing, the phosphopeptides were eluted with 10% ammonia and desalted with C18 ZipTips.

## LC-MS/MS analysis

Phosphorylated peptides were separated using a NanoElute ultra-high-performance liquid system. The capillary ion source was ionized and analyzed by a timsTOF Pro (Bruker) mass spectrometer. For the data shown in Fig. 5C, samples were passed through an ACE C4 column (Avantor). Mass spectrometry was conducted on an AB SCIEX Triple TOF™ 4600 (Foster City), a hybrid triple quadrupole time-of-flight mass spectrometer equipped with a Duo Spray Ion Source. For the data shown in Fig. 5D, Myl12a and phosphorylated Myl12a were analyzed using a 12% NuPAGE gel. Then, the bands were cut from the gel, and the gel samples were digested with trypsin as described previously (Yang et al, 2020).

The tryptic peptides were loaded onto a homemade reversed-phase analytical column and subjected to an NSI source followed by tandem mass spectrometry (MS/MS) in Q ExactiveTM Plus (Thermo).

## Protein crystallization

The concentration of SteC 202-375-C276S used for crystallization was 5 mg/ml. Final concentrations of 2.5 mM APPCP and 5 mM MgCl₂ were added to the SteC 202-375-C276S protein. Screening of crystals was carried out using a droplet vapor diffusion method at 18 °C. The crystals of SteC-APPCP-Mg²⁺ were obtained in a buffer containing 10 mM magnesium sulfate heptahydrate, 50 mM Sodium carbonate trihydrate pH 6.5, and 2.0 M ammonium sulfate after 2 weeks.

## Structure determination

The diffraction data of SteC-APPCP-Mg²⁺ were collected from the BL02U1 line station of a synchrotron radiation light source in Shanghai, China, and the data were processed by the XDS system. The structure of SteC-APPCP-Mg²⁺ was determined using the anomalous scattering method using IPCAS2.0 SAD mode (Ding et al, 2020). In IPCAS2.0 SAD mode, shelxCD, OASIS, phenix.-autobuild, and buccaneer are called to build the initial model, and the subsequent models are manually built using PHENIX and COOT. The structural diagram is completed by PyMol.

## Electron microscopy

RAW264.7 cells infected with *Salmonella* for 10 h were analyzed by electron microscopy. The cells were collected and centrifuged after being treated with an electron microscope-fixing solution. Then, 1% agarose solution was added for pre-embedding. The samples were fixed with 1% osmium tetroxide at room temperature for 2 h and washed twice with PBS. After dehydration with ethanol and acetone, the samples were embedded, permeated, and dried overnight at 37 °C, and polymerized for 48 h at 60 °C. Thin slices were obtained using a Leica UC7 ultra-thin slicer, and then stained and placed on a 150-mesh copper grid. The copper grid was placed in a copper grid box and air-dried overnight in a dark room. Images were obtained using an HT7800/HT7700 (Hitachi) transmission electron microscope.

## Molecular docking

To study the binding of SteC to ATP and UTP, Autodock Tools 1.5.6 was used with a flexible ATP or UTP molecule and a rigid SteC structure. The number of grid points in the XYZ of the grid box was set to 50 × 50 × 50, the grid spacing was 0.375 Å, the number of GA run was set to 200, and default settings were used for the remaining parameters. Finally, the structure with the lowest docking energy was improved with energy minimization. Application of an Amber 14SB force field was used to obtain the appropriate conformations and binding positions of ATP or UTP by energy optimization. This optimization process was carried out in two steps, using a steepest descent method optimization of 2000 steps, followed by optimization of this model by 2000 steps with the conjugate gradient method.

To study the interaction between SteC 168-375 aa and Myl12a, the rigid-body docking program Zdock (ZDOCK3.0.2) was used to predict complex structures. The predicted structure with the highest ZDOCK score was used for further optimization. The final ZDOCK score of the SteC-Myl12a model was 1208.65. The energy optimization of the model was performed as described above.

## Molecular dynamics simulation

The MD simulations of SteC to ATP and UTP models were performed using the Gromacs 2019.6 program under constant temperature and pressure conditions. The Amber 14SB all-atom force field and OPC3 water model were applied. In the MD simulation, all hydrogen bonds were restrained using the LINCS algorithm, and the integration time step was 2 fs. The electrostatic interactions were calculated using the Particle-mesh Ewald (PME) method. The non-bonded interactions were truncated at 10 Å and updated every 10 steps. The temperature of the simulation was controlled at 300 K using the V-rescale temperature coupling method, and the pressure was maintained at 1 bar using the Parrinello-Rahman method. First, the steepest descent method was used to minimize the energy of the two systems to eliminate the atomic contacts. Then, 100 ps of NVT equilibrium simulation was performed at 300 K. Finally, 100 ns of MD simulation was carried out for each system, and the conformation was saved every 20 ps. The visualization of the simulation results was completed using built-in programs in Gromacs and VMD. The MD simulation of the SteC-Myl12a model was conducted using the Gromacs 2018.4 program as described above.

## Phylogenetic analyses and sequence alignment

Amino acid sequences of SteC homologous were obtained from the NCBI database using PSI-blastp. Homologous regions were aligned using MUSCLE and alignments were adjusted manually using the BioEdit sequence alignment editor. Phylogenetic trees were generated using maximum-likelihood methods with bootstrap analysis ($n = 1000$) using FastTree software. Trees were drawn and modified by iTOL.

## In vitro vascular penetration assay

Huvec cells were inoculated into the transwell chamber at the rate of $2 \times 10^6$ per well. After overnight culture, the cells were fixed with 4% paraformaldehyde for 2 h, the lower part of the cell was gently rubbed with a sterile cotton swab, and the part of the cell was cleaned with PBS five to eight times. Afterward, $2 \times 10^5$ RAW264.7 cells (either infected or uninfected) were seeded onto the cells. Fixed staining counts were conducted after 24 h of culturing. For Fig. 7L, pSteC cells were subsequently seeded into the Huvec cell-fixed chambers and placed onto a confocal dish for imaging with a confocal microscope. The cells inside the chamber were real-time imaged using a Zeiss LSM 980 confocal microscope.

## Data availability

The X-ray structure (coordinates and structure factor files) of SteC has been submitted to PDB under accession number 8JBI. Mass spectrometry-based data were submitted to ProteomeXchange under accession number PXD042038.

## Peer review information

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

## Acknowledgements

We thank Prof. Hua Tang and Prof. Jie Yan for their helpful discussion and suggestions. We acknowledge the staff at beamline BL02U1 of the Shanghai Synchrotron Radiation facility for supporting data collection of protein crystals. We acknowledge Dr. Wenyi Du (Sichuan Model Technology Co., Ltd.) for his help in molecular simulations. This work was supported by the National Natural Science Foundation of China [32170034, 22107059, and 32300019], the Taishan Scholar Project of Shandong Province [tsqn202211216 and tsqn202211221], the Natural Science Foundation of Shandong Province [ZR2023YQ060, ZR2022YQ66, and ZR2023QC221], Academic promotion program of Shandong First Medical University [2019LJ001].

## Author contributions

**Yuanji Dai**: Data curation; Investigation; Writing—review and editing. **Min Zhang**: Data curation; Investigation; Writing—review and editing. **Xiaoyu Liu**: Data curation; Investigation; Writing—review and editing. **Ting Sun**: Data curation; Investigation; Writing—review and editing. **Wenqi Qi**: Investigation. **Wei Ding**: Investigation; Writing—review and editing. **Zhe Chen**: Investigation; Writing—review and editing. **Ping Zhang**: Investigation; Writing—review and editing. **Ruirui Liu**: Investigation; Writing—review and editing. **Huimin Chen**: Investigation; Writing—review and editing. **Siyan Chen**: Investigation; Writing—review and editing. **Yuzhen Wang**: Investigation; Writing—review and editing. **Yingying Yue**: Investigation; Writing—review and editing. **Nannan Song**: Investigation; Writing—review and editing. **Weiwei Wang**: Investigation; Writing—review and editing. **Haihong Jia**: Investigation; Writing—review and editing. **Zhongrui Ma**: Investigation; Writing—review and editing. **Cuiling Li**: Investigation; Writing—review and editing. **Qixin Chen**: Funding acquisition; Investigation; Writing—review and editing. **Bingqing Li**: Supervision; Funding acquisition; Investigation; Writing—original draft; Project administration; Writing—review and editing.

## Disclosure and competing interests statement

The authors declare no competing interests.

