## [Peer Review File · The EMBO Journal]

Salmonella manipulates macrophage migration via SteC-mediated myosin light chain activation to penetrate the gut-vascular barrier

Yuanji Dai, Min Zhang, Xiaoyu Liu, Ting Sun, Wenqi Qi, Wei Ding, Zhe Chen, Ping Zhang, Ruirui Liu, Huimin Chen, Siyan Chen, Yuzhen Wang, Yingying Yue, Nannan Song, Weiwei Wang, Haihong Jia, Zhongrui Ma, Cuiling Li, Qixin Chen, and Bingqing Li

Corresponding author(s): Bingqing Li (bingqingsdu@163.com) , Qixin Chen (chenqixin@sdfmu.edu.cn)

Review Timeline:

Submission Date:	30th Jul 23
Editorial Decision:	26th Sep 23
Appeal Received:	1st Feb 24
Editorial Decision:	21st Feb 24
Revision Received:	24th Feb 24
Accepted:	5th Mar 24

Editor: Ieva Gailite

Transaction Report:

Dear Dr. Li,

Thank you for submitting your manuscript for consideration by The EMBO Journal. We have now received two reviewer reports on your manuscript, which are included below for your information. Since the third reviewer was not able to provide their report within a reasonable timeframe, I am taking the decision based on the comments at hand to avoid further delays. Based on the submitted comments, we unfortunately had to conclude that the study is not a sufficiently strong candidate for publication in The EMBO Journal.

As you can see, while both reviewers find the presented structural and in vitro data of interest, they also indicate multiple concerns regarding the methodological approach and the analysis that affect the core conclusions of the manuscript regarding the in cellulo and in vivo function of the Salmonella effector SteC. They also find that a number of important controls have been omitted, thus impacting the conclusiveness of the findings. Given these critical opinions from good experts in the research field, and since a major experimental revision beyond our usual 3-month timeframe and with an uncertain outcome would be needed to address the main referee concerns, I am afraid that we cannot offer further consideration of the manuscript at The EMBO Journal.

Thank you in any case for the opportunity to consider this manuscript. I regret that I could not communicate more positive news, but I nevertheless hope that you will find our reviewers' comments helpful for further improvement of the manuscript.

Yours sincerely,

Ieva Gailite

Referee #1:

In this paper, Dai et al suggest a role for the Salmonella effector, SteC, in macrophage migration and bacterial dissemination. Using infected RAW 264.7 cells and stable cell lines, transwell assays reveal a role for SteC in promoting cell migration and invasion in a manner dependent on its kinase activity. The authors performed a yeast two-hybrid assay to reveal Myl12a as an interactor/substrate of SteC, which was subsequently validated through recombinant protein assays and mass spectrometry. Determination of the SteC crystal structure led to novel insight regarding its NTP usage for kinase activity. Lastly, the authors perform in vivo bioluminescence to demonstrate SteC is necessary systemic spread, presumably through breaching the gut-vasculature barrier.

With exception of the in vitro protein related work (Figure 4, 5, 6), we feel the authors draw overly bold conclusions based on limited amounts of in vitro and in vivo data as outlined in the major comments found below.

Major Points

-All in vitro experiments were performed with RAW264.7 cells. To say this is representative of all macrophages is inadequate, especially without testing migration/invasion in primary cells (i.e. bone marrow-derived macrophages).

-For many experiments there seems to be a lack of controls to confidently interpret the data. Examples of this include:

1) Flow cytometry: the absence of a viability dye to ensure the overall health of GFP+ cells (Figure 1B).

2) Imaging Primary Cells: In Figure 1H, the image containing GFP+ Salmonella is not very compelling, and it appears there is a lot of GFP auto-fluorescence. It would be more convincing to stain these cells against Salmonella (i.e. antiserum) alongside the GFP signal. This is also important to discern how many GFP+ bacteria are within each cell...from this image it looks like a lot of bacteria per cell but seems unlikely given these cells are isolated only one hour post-infection.

3) Migration: While a time-lapse experiment is indeed informative (Figure 2A), the lack of a control (i.e. uninfected cells) weakens the argument. If the uninfected cells within each frame are the "controls", then quantitative analysis is required to ensure this is not just specific to the single field of view provided. Considering Figure 1A emphasizes movement within 80

minutes, why was migration/invasiveness not investigated at another timepoint (i.e. 24 hours?). Keeping in mind that RAW cells replicate ~ every 15 hours, perhaps infected cells replicating on the coverslip overestimate the extent of migration. In Figure 2E, a positive control for macrophage migration (e.g. cytokine) should be included to gauge how much a virulence factor can promote migration relative to a host protein - especially if this assay is the main readout. In Figure 2H and 2I, how come so few cells are infected despite an MOI=10? I would expect the representative images used in this figure to mimic the conditions tested. Normalization of actin intensity is required (e.g. cell volume/area) to account for the apparent differences in cell shape between WT- and dSteC-infected cells demonstrated in 2H. Furthermore, there is no image of uninfected cells to demonstrate that wildtype-infection is promoting podosome formation.

4) Stable Cell Lines: In Figure 3A, protein validation of SteC expression is required to adequately compare WT and K256H-stable cell lines. Furthermore, confusion arises when a GFP stable cell line is used as a control - this seems inappropriate for a stable cell line which does not have a fluorescent protein itself (Or does SteC include GFP? In which case this was not clear in the text). Importantly, RAW264.7 cells are morphologically diverse (<https://www.atcc.org/products/tib-71>) so a lower magnification, scanning electron microscope image of uninfected and infected cells would be more convincing to suggest that WT-infection induces morphological changes (or the equivalent in the inducible cell lines).

5) Drug Treatments: It seems unlikely that broad drugs such as MLCK and ROCK inhibitors do not impair the function of RAW264.7 cells. Without a dose-response, or positive control that the drugs work (e.g. another marker in RAW cells or even fibroblasts as a control), how is the reader to believe that the SteC-dependent phenotype is independent of the MEK-ERK-MLCK pathway.

6) Myl12a Phosphorylation: While the in vitro work demonstrating SteC-mediated phosphorylation of Myl12a is convincing, the absence of data demonstrating this during Salmonella infection is troublesome. We acknowledge that confirming this in a stable cell line is a step forward but it seems incomplete to not assess this during infection.

7) Bacterial Counts: Insufficient data to conclude differences between the two strains. Assessing this by TEM images is biased and dismisses potential differences in the frequency of infection. Measuring bacterial CFUs overtime or even looking at the GFP intensity across many more cells (than 5) would be sufficient.

8) My12a Knockout Cells: The authors do not provide any confirmation of Myl12a knockout in the cell lines used. It is also important to measure how migration and invasion differs in uninfected WT and KO cells alongside infected (Figure 7C).

9) In vivo: PV-1 is used as a marker of impaired GVB but the type of impairment is unclear and only recently published (Bertocchi et al., 2021). More importantly, the lack of PV-1 staining in uninfected animals raises skepticism regarding how staining of this protein is supposed to look - let alone between mice infected with different bacterial strains. Lastly, it is unclear where Salmonella is localized in SteC knockout mice; it seems incomplete to only include images in WT-infected animals. Staining for immune cells would also be insightful to see if there are differences in recruitment within the gut.

Minor Points

-Figure 4D: Scale bar is missing. Insets are required alongside quantitative analysis (co-localization analysis) to entice the idea that the puncta are overlapping. Alternative proof would include a pulldown experiment from co-transfected cells as this would complement recombinant protein data in 4E.

Referee #2:

This manuscript presents evidence that the secreted Salmonella effector protein SteC acts as a kinase to influence the motility of infected macrophages, both in vitro and in vivo. SteC is secreted by the Salmonella T3SS-2 secretion system, which is activated after infection of the host cell. The kinase activity of SteC has been reported previously by others, who also showed that it promotes actin cytoskeletal rearrangements in fibroblasts. Here the authors examine the activity of SteC in macrophages, where it phosphorylates several myosin light chains to promote motility and invasion, as measured by transwell filter migration assays. In vivo studies of Salmonella spread indicate that a SteC deficient strain is impaired in dissemination from the small intestine to other organs. Based on these observations, the authors conclude that SteC is important in the ability of Salmonella Typhimurium to penetrate the vascular bed and enter the circulation. However, while it is clear that SteC enhances the mobility of infected macrophages, there is no direct evidence in this study that this promotes entry into the vasculature.

The most interesting and novel aspect of this study is the structural determination of SteC, which is clearly a novel form of kinase with homologs that exist in multiple bacterial pathogens. The data indicate that SteC acts as an obligate dimer (in contrast to most other kinases that can act as monomers). SteC is also apparently capable of using any NTP (ATP, GTP, UTP, CTP) as the phosphate donor, which the authors suggest may allow it to function in ATP-depleted cells, as occurs during inflammation.

While this structural and biochemical characterization of SteC is well done and valuable to the field, several other aspects of this study require attention:

1. Fig. 1 probes the presence of GFP-expressing *S. Typhimurium* in the blood of orally infected mice, suggesting that most of the infected cells are F4/80-positive macrophages. Apparently ~70% of GFP+ cells are NOT macrophages, which the authors attribute to background fluorescence. How did they determine this? What was the threshold for "background fluorescence"?

2. It is also surprising that the data in Fig. 1 were acquired only 1h after oral infection. Typically it takes several hours for gavaged bacteria to even penetrate the intestinal epithelium, let alone reach the circulation. Considering that most of their other in vivo data was acquired at least 10h after infection, why did the authors choose such a short time point for the analysis in Fig. 1?
3. The images shown in Fig. 1H are of poor quality. There is a diffuse green signal in the entire cell, and bacteria are barely detectable. Better quality, higher magnification images are essential here to demonstrate the presence of intracellular bacteria.
4. The finding that migration and invasion of Matrigel are completely blocked by the actin depolymerizing agent latrunculin B (Fig. 2F) is not surprising. Also, the images shown in Fig. 2H are too small; bacteria are barely visible. The quantitation shown in Fig. 2I represents actin fluorescence intensity, yet the changes in actin organization are impossible to discern in the corresponding images. The authors claim that infected cells form podosomes, but these are not detectable in the images shown. How do the authors define podosomes? What other markers of podosomes were used?
5. In Fig. 3 the authors generated stable macrophage cell lines (RAW264.7) expressing either WT SteC or a catalytically inactive mutant (K256H). Although the data indicate a strong dependence of motility and invasion on SteC catalytic activity, the authors need to demonstrate that the WT and mutant proteins are expressed at equivalent levels in the two lines.
6. The concentrations of the inhibitors used in Fig. 3F-I should be provided.
7. The colocalization of SteC with Myl12a (Fig. 4D) needs to be quantified across at least 20 cells. What do the authors think the highlighted puncta represent? If they think these are podosomes, a corresponding podosome marker should be included in the images.
8. Fig. 5 shows convincingly that SteC phosphorylates Myl12a in vitro and in cells overexpressing SteC. Is Myl12a phosphorylated in cells infected with *S. Typhimurium*?
9. A table containing the results of the phosphoproteomic screen should be provided in support of Fig. 5H.
10. As noted above, the authors conclusion that SteC is required for breaching the intestinal vascular barrier is an over-interpretation of the data. The images shown in Fig. 7G and 7H have several problems. First, they are too small, and bacteria are barely detectable. Second, an earlier time point should be shown at which both WT and Δ SteC bacteria are detectable in the lamina propria. Third, images acquired from multiple sections from at least 3 different mice need to be quantified to be convincing.
11. Finally, there is no evidence in these images that Δ SteC bacteria are incapable of penetrating the vasculature. Their presence in the lamina propria at this single time point could easily represent a failure to migrate away from infected epithelial cells rather than an inability to penetrate the vascular basement membrane and endothelial layer.
12. In their discussion the authors offhandedly refer to an observation that Δ SteC strains exhibit a higher toxicity relative to WT strains. What do they mean by "toxicity" and in what context? What kind of cells? Do the Δ SteC strains kill their host macrophages?

** As a service to authors, EMBO Press provides authors with the possibility to transfer a manuscript that one journal cannot offer to publish to another EMBO publication or the open access journal Life Science Alliance launched in partnership between EMBO Press, Rockefeller University Press and Cold Spring Harbor Laboratory Press. The full manuscript and if applicable, reviewers' reports, are automatically sent to the receiving journal to allow for fast handling and a prompt decision on your manuscript. For more details of this service, and to transfer your manuscript please click on Link Not Available. **

Manuscript EMBOJ-2023-115180

Response to the referees' comments

We thank to the Editor and Referees for their valuable commentary and constructive suggestions. We believe that the revised manuscript, which has been extensively expanded, has improved significantly as a result. We hope that all parties involved find our revision efforts persuasive and look forward to finalizing our study for publication in the EMBO Journal.

The main changes made in this manuscript were:

- New data revealing that *Salmonella* accumulates in macrophages surrounding the blood vessels of small intestine during the early infection stages (**Fig.1A-D**).
- Real-time cell movement trajectory analysis was applied to further investigate the migration phenotype of macrophages following infection (**Fig.2E and Fig.S1**).
- New data demonstrating *Salmonella* promotes the migration/invasion phenotype of bone marrow-derived macrophages (BMDM) (**Fig.2E-F**).
- New data confirming Myl12a undergoes SteC-dependent phosphorylation in both RAW264.7 and BMDM macrophages during *Salmonella* infection (**Fig.5G-I**).
- Direct evidence supporting *Salmonella*-infected macrophages could traverse the vascular epithelial barrier (**Fig.7J-L**).

In addition, the revised manuscript incorporated the suggested controls based on the feedback from the reviewers. In the revised manuscript, the quality of the immunofluorescence imaging results (**Fig.1L and Fig.7H**) was significantly improved. We believe the revised manuscript has adequately addressed all the major concerns of the reviewers. Below, we present a point-by-point response to each of the referees' comments.

Response to Referee #1:

In this paper, Dai et al suggest a role for the Salmonella effector, SteC, in macrophage migration and bacterial dissemination. Using infected RAW 264.7 cells and stable cell lines, transwell assays reveal a role for SteC in promoting cell migration and invasion in a manner dependent on its kinase activity. The authors performed a yeast two-hybrid assay to reveal Myl12a as an interactor/substrate of SteC, which was subsequently validated through recombinant protein assays and mass spectrometry. Determination of the SteC crystal structure led to novel insight regarding its NTP usage for kinase activity. Lastly, the authors perform in vivo bioluminescence to demonstrate SteC is necessary systemic spread, presumably through breaching the gut-vasculature barrier.

With exception of the in vitro protein related work (Figure 4, 5, 6), we feel the authors draw overly bold conclusions based on limited amounts of in vitro and in vivo data as outlined in the major comments found below.

Response: Thank you very much for your constructive feedback. During the revision process, based on your suggestions and those of another reviewer, we have included a significant amount of *in vivo* and *in vitro* experimental data. The main changes made in this manuscript were:

- New data revealing that *Salmonella* accumulates in macrophages surrounding the blood vessels of small intestine during the early infection stages (**Fig.1A-D**).
- Real-time cell movement trajectory analysis was applied to further investigate the migration phenotype of macrophages following infection (**Fig.2E and Fig.S1**).
- New data demonstrating *Salmonella* promotes the migration/invasion phenotype of bone marrow-derived macrophages (BMDM) (**Fig.2E-F**).
- New data confirming Myl12a undergoes SteC-dependent phosphorylation in both RAW264.7 and BMDM macrophages during *Salmonella* infection (**Fig.5G-I**).
- Direct evidence supporting *Salmonella*-infected macrophages could traverse the vascular epithelial barrier (**Fig.7J-L**).

In the revised version, Figure 1, Figure 2, Figure 5, and Figure 7 have all been significantly improved compared to the previous version. We believe that the current version better supports our conclusions and kindly submit it for your review.

Major Points

-All in vitro experiments were performed with RAW264.7 cells. To say this is representative of all macrophages is inadequate, especially without testing migration/invasion in primary cells (i.e. bone marrow-derived macrophages).

Response: Thank you very much for your valuable suggestions. We conducted a series of experiments using primary bone marrow-derived macrophages (BMDM) to assess their motility and invasiveness after *Salmonella* infection. As illustrated in **Figure 1A-C** below, *Salmonella* infection resulted in a significantly enhancement of

motility and invasiveness in BMDM cells. However, the $\Delta steC$ strain did not induce invasive phenotype of BMDMs, providing further evidence that *Salmonella*-driven migration/invasion properties of macrophage is dependent on SteC. Following your advice, we also investigated the phosphorylation of Myl12a in BMDM cells after *Salmonella* infection. The results showed that phosphorylation of Myl12a was detected as early as 15 minutes after wild-type *Salmonella* infection, whereas no corresponding phosphorylation was observed in the $\Delta steC$ strain (Figure 1E-F). These data strongly supports our conclusion that during *Salmonella* infection of primary macrophages and RAW264.7 cells, SteC catalyzes the phosphorylation of Myl12a, thereby activating cellular motility and invasiveness. The aforementioned data has been included in Figures 2 and 5 of the revised version.

Figure 1. *Salmonella*-induced migration/invasion properties of BMDMs.

A. Movement trajectories of BMDMs before and after *Salmonella* infection.

B. Presentation of individual cell movement trajectories.

C-D. Transwell results of BMDMs after *Salmonella* infection.

E-F. Rapid activation of Myl12a phosphorylation in BMDM after infection

-For many experiments there seems to be a lack of controls to confidentially interpret the data. Examples of this include:

1) Flow cytometry: the absence of a viability dye to ensure the overall health of GFP+ cells (Figure 1B).

Response: Thank you for your question. In our flow cytometry experiments, we employed Zombie NIR (Biolegend, 423105) labeling to discriminate and assess the viability of cells. This was done to facilitate subsequent analysis (**Figure 2**). We sincerely apologize for the omission of this pertinent information in the previous version, and we have rectified the oversight by incorporating the details of this analysis process in the new version (line 915-916).

Figure 2. Cells were stained with vitality dye Zombie NIR to identify live leukocytes by flow cytometry.

2) Imaging Primary Cells: In Figure 1H, the image containing GFP+ Salmonella is not very compelling, and it appears there is a lot of GFP auto-fluorescence. It would be more convincing to stain these cells against Salmonella (i.e. antiserum) alongside the GFP signal. This is also important to discern how many GFP+ bacteria are within each cell...from this image it looks like a lot of bacteria per cell but seems

unlikely given these cells are isolated only one hour post-infection.

Response: Thank you for your kind suggestions. The previous imaging results were problematic due to several identified factors. Firstly, we discovered that the weak specificity of the GFP antibody resulted in non-specific staining, leading to inaccurate results. Secondly, there was a decrease in the number of viable cells available for analysis during the flow cytometry process (~4h), possibly due to cell death or lysis. To overcome these challenges, we conducted additional experiments. Specifically, we directly employed isolated blood for the immunofluorescence assay in order to minimize cell death and enhance the quality of the experimental procedure. Additionally, *Salmonella*-specific antibody (abcam ab35156) was used. These interventions proved successful in obtaining clearer imaging of F4/80-positive macrophages containing *Salmonella*. Importantly, each observed cell demonstrated the presence of 1-2 bacteria, as depicted in **Figure 3**. This improved imaging provides a more precise and reliable representation of the experimental findings. Consequently, we have included this figure as part of the revised version, specifically in **Figure 1L**.

Figure 3. F4/80-positive macrophages containing *Salmonella* were isolated from mouse blood after a 1-hour infection.

3) *Migration:* While a time-lapse experiment is indeed informative (Figure 2A), the lack of a control (i.e. uninfected cells) weakens the argument. If the uninfected cells within each frame are the "controls", then quantitative analysis is required to ensure this is not just specific to the single field of view provided.

Response: We apologize for the oversight of not including an uninfected control in our previous experiment. By consulting relevant literature, we performed new experiments to evaluate the motility of cells before and after infection. The cell movement trajectories of each cell were analyzed by Imaris software (**Figure 4**). The results clearly demonstrated that infection with wild-type *Salmonella* strain indeed activated cell mobility of Raw264.7, whereas infection with Δ *steC* strain resulted in

the inhibition of cell mobility. These findings are consistent with previous reports suggesting that antigen-activated macrophages exhibit reduced mobility.

Figure 4. The movement trajectories of Raw264.7 cells before or after WT/ Δ steC infection.

Considering Figure 1A emphasizes movement within 80 minutes, why was migration/invasiveness not investigated at another timepoint (i.e. 24 hours?). Keeping in mind that RAW cells replicate ~ every 15 hours, perhaps infected cells replicating on the coverslip overestimate the extent of migration.

Response: Thank you for your valuable suggestions and guidance. Initially, we chose to adopt a 24-hour time point for our transwell experiment, following some references about motility of Raw264.7 (PMID: 26202362, PMID: 19241443). Upon considering your advice, we conducted an additional transwell experiment using BMDM cells and performed a time gradient analysis at 2-4 hours post-infection. Our observations revealed a significant increase in the mobility and migration of BMDMs after infection. To further investigate the underlying mechanisms, we conducted phosphorylation experiments on Myl12a, which revealed that Myl12a phosphorylation initiated as early as 15 minutes after *Salmonella* infection. **Therefore, the induction of macrophage mobility by *Salmonella* occurs in the early stages of infection.** To analyze whether SteC expression affects the proliferation of Raw264.7 cells, we conducted an analysis of cell proliferation in pSteC and pGFP. Our findings indicate that SteC expression has minimal impact on cell proliferation (**Figure 5**). This rules out the possibility of false positives resulting from differences in replication rates.

Figure 5. The cell proliferation analysis of pSteC and pGFP.

In Figure 2E, a positive control for macrophage migration (e.g. cytokine) should be included to gauge how much a virulence factor can promote migration relative to a host protein - especially if this assay is the main readout.

Response: Thank you for your valuable suggestions. Based on your suggestion and literature review, we included CCL2 and CCL7 as positive controls in our study, as presented in **Figure 6**. It was observed that CCL2 significantly stimulated macrophage mobility and invasion, while CCL7 had minimal effect. Comparing these results with the mobility and invasive capabilities induced by *Salmonella* infection, we found that *Salmonella* infection exhibits a similar effect to promote motility and invasion with CCL2. This finding provides important support for the concept that *Salmonella* infection triggers macrophages to cross the vascular barrier. We have included this data in **Fig.2C** of the main text.

Figure 6. Comparison of the transwell results between cytokines and *Salmonella* infection.

In Figure 2H and 2I, how come so few cells are infected despite an MOI=10?

Response: Thank you for your query. In this experiment, we initially infected cells with a multiplicity of infection (MOI) of 10 for 1 hour. Following this, we eliminated the extracellular bacteria in order to observe the infected bacteria. It is possible that the limited resolution of the images we provided may have obscured the clear visibility of bacterial presence emitting green fluorescence within the cells of the majority of the infected group. In fact, more than half of infected cells exhibited green fluorescent bacteria (**Figure 7**). It should be noted that due to the three-dimensional nature of RAW cells, it is possible that some cells did not display visible green fluorescence despite the presence of internalized bacteria. For instance, in the dynamic imaging experiment depicted in **Figure 8**, it can be observed that although cells 1 and 2 contained bacteria, this may not have been visible in certain images due to the position of the fluorescent signal (Time point 1 and Time point 4). We hope that this clarification addresses your concerns adequately.

Figure 7. Half of infected cells exhibited green fluorescent bacteria.

Figure 8. At certain times and orientations, fluorescent labeled bacteria inside the cell were not captured.

I would expect the representative images used in this figure to mimic the conditions tested.

Response: Thank you for your suggestion. We have included representative images (**Figure 9**) in the new version.

Figure 9. Representative images of *Salmonella* induced actin rearrangement

Normalization of actin intensity is required (e.g. cell volume/area) to account for the apparent differences in cell shape between WT- and Δ SteC-infected cells demonstrated in 2H.

Response: Thank you for your suggestion. It is possible that a detailed description of the statistical method was not provided, however, it should be noted that the calculation of actin intensity has been involved averaging the area. The raw data and data processing procedures have been uploaded to Mendeley Data (DOI:10.17632/t7k6g6sns3.2, Figure 2 and Figure 3). **Figure 10** illustrates representative picture of processed data. Further elaboration on the method can be found in the corresponding section (line 907-912).

Figure 10. Some original graphs of statistical data for normalization of actin intensity.

Furthermore, there is no image of uninfected cells to demonstrate that wildtype-infection is promoting podosome formation.

Response: Thank you for your suggestion. We have included results from the uninfected control group in our analysis (**Figure 9**).

4) Stable Cell Lines: In Figure 3A, protein validation of SteC expression is required to adequately compare WT and K256H-stable cell lines.

Response: Thank you for your suggestion. We have used the SteC antibody (Dia-An Biotech, Custom made) to demonstrate that there is no difference in the expression of the SteC protein between the wild type and mutant strains (**Figure 11**). We have included these results in **Fig.3A** of the revised manuscript.

Figure 11. Protein validation of SteC expression in WT and K256H-stable cells

Furthermore, confusion arises when a GFP stable cell line is used as a control - this seems inappropriate for a stable cell line which does not have a fluorescent protein itself (Or does SteC include GFP? In which case this was not clear in the text).

Response: Thank you for your question. We have included a statement in the manuscript explaining that the SteC protein in our constructed plasmid is not fused with GFP (line 203-204). Therefore, we consider cells with stable GFP expression to serve as a negative control. The plasmid maps are depicted below (**Figure 12**).

Figure 12. Plasmid maps of SteC expression stable cell lines

Importantly, RAW264.7 cells are morphologically diverse (<https://www.atcc.org/products/tib-71>) so a lower magnification, scanning electron microscope image of uninfected and infected cells would be more convincing to suggest that WT-infection induces morphological changes (or the equivalent in the inducible cell lines).

Response: Thank you for your question. It is indeed well-known that RAW264.7 cells can exhibit heterogeneous morphologies under certain culture conditions. However, under our experimental conditions, non-infected RAW264.7 cells maintain a uniform spherical shape, with a differentiation rate below 5%. **Figure 13** represents images captured during the cultivation of RAW264.7 cells. In order to clearly demonstrate the morphological changes before and after infection, we have included bright-field images (**Figure 9**). These images vividly depict the induced morphological changes in RAW264.7 cells following wild-type *Salmonella* infection, while no such changes are observed in the knockout strain. Given that morphological changes are not the focus of our study, we did not conduct scanning electron microscopy experiments to further examine the cellular morphology. We hope for your understanding in this regard.

Figure 13. The morphology of Raw264.7 cells cultivated in our laboratory.

5) Drug Treatments: It seems unlikely that broad drugs such as MLCK and ROCK inhibitors do not impair the function of RAW264.7 cells. Without a dose-response, or positive control that the drugs work (e.g. another marker in RAW cells or even fibroblasts as a control), how is the reader to believe that the SteC-dependent phenotype is independent of the MEK-ERK-MLCK pathway.

Response: Thank you for your question. The MLCK inhibitor used in this study is ML-7 (product number: MCE, HY-15417). ML-7 has an IC₅₀ of 300 nM for inhibiting MLCK activity. Existing literature shows that ML-7 hydrochloride at concentrations of 3 μ M and 10 μ M can effectively inhibit the activity of myosin light chain kinase (MLCK) and subsequently attenuate contraction induced by Dexmedetomidine (DMT) (PMID: 26392810). Therefore, the concentration of 10 μ M used in our study can theoretically result in significant inhibition of MLCK activity. To confirm the inhibitory effect of ML-7 on MLCK activation in our research system, we performed the following experiments. Initially, we activated MLCK in raw264.7 cells using MLCK activators (16-38-Thymosin β 4, MCE, HY-P3965). Subsequently, we added ML-7 inhibitors to the cells to assess the phosphorylation of Myl12A. The results demonstrated that the addition of 5-10 μ M ML-7 significantly inhibited the MLCK-induced phosphorylation of MYL12A (**Figure 14A**). This finding substantiates the effectiveness of the ROCK inhibitor concentration employed in our study. Y-33075 (also called Y-39983) is a potent ROCK inhibitor, with an IC₅₀ of 3.6 nM. Y-33075 (10 μ M) extends neurites in the retinal ganglion cells (RGCs) compared with those in RGCs treated without Y-33075 (PMID: 21950703). Y-33075 (1 μ M) inhibits the contraction of rabbit ciliary artery segments evoked by histamine in Ca²⁺-free solutions (PMID: 21667088). Therefore, the concentration of 10 μ M used in our study can theoretically result in significant inhibition of ROCK activity. We conducted a similar experiment using Y-33075 at a concentration of 10 μ M. The results showed that Y-33075 effectively inhibited the phosphorylation of Myl12A induced by ROCK activator (Pentanoic acid, HY-N6056) at this concentration

(Figure 14B). Above findings confirm that the inhibitor concentration we used in our study was effective.

Figure 14. Evaluation of the inhibitory efficacy of MLCK and ROCK inhibitors

6) *Myl12a* Phosphorylation: While the *in vitro* work demonstrating *SteC*-mediated phosphorylation of *Myl12a* is convincing, the absence of data demonstrating this during *Salmonella* infection is troublesome. We acknowledge that confirming this in a stable cell line is a step forward but it seems incomplete to not assess this during infection.

Response: That is a valuable suggestion. Consequently, we conducted separate infection experiments using RAW264.7 and BMDM cells to assess the phosphorylation levels of *Myl12a* during *Salmonella* infection. Remarkably, the results showed that phosphorylation of *Myl12a* was evident in WT-infected cells, whereas no phosphorylation of *Myl12a* was detected in the Δ *steC* infected cells (Figure 15). These findings strongly indicate that *Salmonella* activates *SteC*-dependent phosphorylation of *Myl12a* during the course of actual infection. The above data has been added to Fig.5 of the revised manuscript.

Figure 15. *Salmonella* infection promotes phosphorylation of Myl12a in a SteC-dependent manner.

7) *Bacterial Counts: Insufficient data to conclude differences between the two strains. Assessing this by TEM images is biased and dismisses potential differences in the frequency of infection. Measuring bacterial CFUs overtime or even looking at the GFP intensity across many more cells (than 5) would be sufficient.*

Response: Thank you for your suggestions. We compared the survival rates of wild-type and knockout bacteria in raw264.7 cells using the CFU counting method (Figure 16). The results indicate that the invasion ability of $\Delta steC$ into host cells was slightly lower than that of WT strain. On the contrary, the survival ability of $\Delta steC$ within host cells was significantly greater compared to the wild-type strain, which is consistent with findings reported in 2012 (PMID: 23159055, Fig.6B). Due to our primary focus is to investigate the regulatory effect of SteC on macrophage motility, and therefore we will not extensively explore the aforementioned phenotypes. The data mentioned were included in Fig.7A. Thank you for your understanding.

Figure 16. The number of intracellular bacteria following infection.

8) *My12a* Knockout Cells: The authors do not provide any confirmation of *My12a* knockout in the cell lines used.

Response: The $\Delta my12a$ cells were generated in collaboration with UBIGENE Company. The confirmation report for the knockout cells is presented below (**Figure 17**). Cell clone #B06 was subjected to sequencing, resulting in the confirmation of the complete knockout of *my12a*. This cell clone was employed in our corresponding experiments. The PCR validation results were incorporated into **Fig.S3** of the revised manuscript.

Figure 17. Confirmation results of *my12a* knockout Raw264.7.

It is also important to measure how migration and invasion differs in uninfected WT and KO cells alongside infected (Figure 7C).

Response: Thank you for your suggestion. We have included results from the uninfected control group in the revised manuscript (**Fig.S3B-C**).

9) *In vivo*: PV-1 is used as a marker of impaired GVB but the type of impairment is unclear and only recently published (Bertocchi et al., 2021). More importantly, the lack of PV-1 staining in uninfected animals raises skepticism regarding how staining of this protein is supposed to look - let alone between mice infected with different bacterial strains.

Response: Thank you for your suggestion. To establish a comparison, we included an uninfected control group (**Figure 18**). Consistent with prior studies, uninfected individuals exhibited lower expression of PV-1. Both WT and $\Delta steC$ infections significantly induced PV-1 expression, with no statistically significant difference observed between the two groups.

Figure 18. PV-1 staining in uninfected and *Salmonella*-infected animals

Lastly, it is unclear where *Salmonella* is localized in *SteC* knockout mice; it seems incomplete to only include images in WT-infected animals. Staining for immune cells would also be insightful to see if there are differences in recruitment within the gut.

Response: Thank you for your suggestion. We made significant improvements to our previous immunofluorescence experiment based on your recommendation. Firstly, we replaced the GFP antibodies with *Salmonella* specific antibodies in order to reduce non-specific signals. Additionally, we conducted immune cell staining and obtained informative results (Figure 19). Our findings indicate that within one hour after infection, *Salmonella*, both WT and $\Delta steC$ strain, have been observed to accumulate in macrophages that surround the microvessels in the villi of the small intestine. After 20 h of infection, $\Delta steC$ still remained within the vascular intestinal macrophages, while WT primarily exists in intestinal epithelial cells. These findings provide strong support for our hypothesis that *SteC* aids *Salmonella* in entering the bloodstream and spreading by facilitating macrophage movement and invasion.

Figure 19. Immunofluorescence analysis revealed the precise localization of *Salmonella* during both early and late stages of infection.

A. Small intestinal (SI) after 1h of *Salmonella* infection were stained with PV-1 (Red), and *Salmonella* (Green). B. Statistics of *Salmonella* distribution in host small intestinal after infection. C. Intestinal macrophages primarily reside around the small intestinal villous microvasculature. D. *Salmonella* primarily colonizes macrophages located in the vicinity of blood vessels after 1h of *Salmonella* infection. E-F. After 20 h of infection, $\Delta steC$ still remained within the vascular intestinal macrophages.

Minor Points

-Figure 4D: Scale bar is missing. Insets are required alongside quantitative analysis (co-localization analysis) to entice the idea that the puncta are overlapping. Alternative proof would include a pulldown experiment from co-transfected cells as this would complement recombinant protein data in 4E.

Response: Thank you for your suggestion. We have incorporated a scale bar of the images. Additionally, we have taken measures to standardize the analysis of co-localization. The colocalization of SteC with Myl12a were quantified across 20 cells as shown in **Figure 20**, the Pearson correlation coefficient (PCC) of each cells were determined.

Figure 20. Co-localization analysis of SteC and Myl12a.

Response to Referee #2:

This manuscript presents evidence that the secreted Salmonella effector protein SteC acts as a kinase to influence the motility of infected macrophages, both in vitro and in vivo. SteC is secreted by the Salmonella T3SS-2 secretion system, which is activated after infection of the host cell. The kinase activity of SteC has been reported previously by others, who also showed that it promotes actin cytoskeletal rearrangements in fibroblasts. Here the authors examine the activity of SteC in macrophages, where it phosphorylates several myosin light chains to promote motility and invasion, as measured by transwell filter migration assays. In vivo studies of Salmonella spread indicate that a SteC deficient strain is impaired in dissemination from the small intestine to other organs. Based on these observations, the authors conclude that SteC is important in the ability of Salmonella Typhimurium to penetrate the vascular bed and enter the circulation. However, while it is clear that SteC enhances the mobility of infected macrophages, there is no direct evidence in this study that this promotes entry into the vasculature.

The most interesting and novel aspect of this study is the structural determination of SteC, which is clearly a novel form of kinase with homologs that exist in multiple bacterial pathogens. The data indicate that SteC acts as an obligate dimer (in contrast to most other kinases that can act as monomers). SteC is also apparently capable of using any NTP (ATP, GTP, UTP, CTP) as the phosphate donor, which the authors suggest may allow it to function in ATP-depleted cells, as occurs during inflammation. While this structural and biochemical characterization of SteC is well done and valuable to the field, several other aspects of this study require attention

Response: Thank you very much for your constructive feedback. During the revision process, based on your suggestions and those of another reviewer, we have included a significant amount of *in vivo* and *in vitro* experimental data. The main changes made in this manuscript were:

- New data revealing that *Salmonella* accumulates in macrophages surrounding the blood vessels of small intestine during the early infection stages (**Fig.1A-D**).
- Real-time cell movement trajectory analysis was applied to further investigate the migration phenotype of macrophages following infection (**Fig.2E** and **Fig.S1**).
- New data demonstrating *Salmonella* promotes the migration/invasion phenotype of bone marrow-derived macrophages (BMDM) (**Fig.2E-F**).
- New data confirming Myl12a undergoes SteC-dependent phosphorylation in both RAW264.7 and BMDM macrophages during *Salmonella* infection (**Fig.5G-I**).
- Direct evidence supporting *Salmonella*-infected macrophages could traverse the vascular epithelial barrier (**Fig.7J-L**).

In the revised version, Figure 1, Figure 2, Figure 5, and Figure 7 have all been significantly improved compared to the previous version. We believe that the current version better supports our conclusions and kindly submit it for your review.

1. Fig. 1 probes the presence of GFP-expressing *S. Typhimurium* in the blood of orally infected mice, suggesting that most of the infected cells are F4/80-positive macrophages. Apparently ~70% of GFP⁺ cells are NOT macrophages, which the authors attribute to background fluorescence. How did they determine this? What was the threshold for "background fluorescence"?

Response: Thank you for your question. In order to study the early bloodstream infections caused by *Salmonella*, animal experiments using GFP-labeled *Salmonella* were conducted with reference to a literature published in 1999 (PMID: 10548107). In the aforementioned study, the authors observed the presence of 0.01% blood cells were GFP positive in the blood of mice that were not infected with GFP labeled *Salmonella*. Since GFP-labeled bacteria were not expected to be present in the bloodstream of these uninfected mice, so these cells were identified as spontaneous fluorescence backgrounds (PMID: 10548107).

During our research, we additionally observed an approximate GFP fluorescence intensity of 0.015% in the blood of uninfected mice, which exceeded our predetermined GFP fluorescence threshold (2.5×10^2). Following flow cytometry sorting, the GFP⁺ cells in blood of uninfected mice were stained and examined, as depicted in the figure below (**Figure 1A**). Notably, GFP-positive cells in the uninfected group were characterized by low fluorescence levels and exhibited fragmented morphology rather than the typical characteristics of macrophages. Based on these findings, we postulate that these GFP⁺ cells can be attributed to background fluorescence of spontaneous nature and should be subtracted as false positive cells. GFP-positive cells were sorted from the infected group for imaging observation, revealing that numerous samples displayed bacterial morphology. Regrettably, the cells harboring these bacteria were fragmented as a result of lysis (**Figure 1A**). We hypothesize that this issue may be attributed to an excessively prolonged duration of flow cytometry sorting (~4h). Therefore, we explored alternative experimental approaches to eliminate the use of flow cytometry sorting. Following a 1-hour infection, we directly isolated mouse blood cells and employed imaging flow cytometry (Amnis ImageStream x Mk II) to identify GFP and F4/80 double-positive cells (**Figure 1B**). Interestingly, the results revealed a distinct population of double-positive cells in the infected group, in contrast to the uninfected control where such cells were rare and displayed weak fluorescence. In order to improve our imaging quality, we employed confocal microscopy and performed staining with *Salmonella*-specific antibodies and F4/80 (**Figure 1C**). The subsequent experimental observations highlighted a significant abundance of *Salmonella*-containing macrophages within the infected group, while these cells were entirely absent in the uninfected control. Furthermore, the weak fluorescence intensity of F4/80 suggested its expression in cell types other than macrophages.

Although it is not possible to eliminate the problem of false positives with GFP, upon subtracting the controls, a significant proportion of GFP-positive cells were found to be positive for F4/80. These results and confocal imaging data supports our conclusion that *Salmonella* enters the bloodstream alongside macrophages within one hour of infection.

Figure 1 Comparison of GFP fluorescence between uninfected control and *Salmonella*-infected groups using various experimental methods.

- A. The immunofluorescence imaging results were obtained after sorting the GFP-positive cells using flow cytometry.
- B. The imaging flow cytometry experiment was performed on peripheral blood cells that exhibited double positivity for EGFP and F4/80.
- C. The confocal imaging results of double-positive for *Salmonella* and F4/80 cells from peripheral blood cells.

2. *It is also surprising that the data in Fig. 1 were acquired only 1h after oral infection. Typically it takes several hours for gavaged bacteria to even penetrate the intestinal epithelium, let alone reach the circulation. Considering that most of their other in vivo data was acquired at least 10h after infection, why did the authors choose such a short time point for the analysis in Fig. 1?*

Response: The phenomenon of *Salmonella* entering the bloodstream shortly after infection was first discovered in 1999 (PMID: 10548107). This article reported that *Salmonella* could enter the bloodstream within 15 minutes post oral infection. Currently, this article has been cited nearly 900 times. Multiple research groups have repeated the above results, all indicating that *Salmonella* can enter the bloodstream within 1 hour after infection (PMID: 17095609, PMID: 23028876, PMID: 37999606). In our study, we discovered that *Salmonella* can indeed be detected in the bloodstream of infected animals as soon as one hour after oral inoculation. This finding was indeed surprising and captivating, prompting our curiosity in exploring the underlying mechanism of this phenomenon. According to the findings in this study, We have identified a plausible explanation that *Salmonella* has the capability to phosphorylates host Myl12a through its effector SteC as early as 15 minutes after infection. Consequently, infected macrophages possess the capacity to migrate and invade surrounding tissues, ultimately disseminating the infection back into the bloodstream. In this manner, macrophages function as Trojan horses, facilitating the spread of *Salmonella* infection.

3. *The images shown in Fig. 1H are of poor quality. There is a diffuse green signal in the entire cell, and bacteria are barely detectable. Better quality, higher magnification images are essential here to demonstrate the presence of intracellular bacteria.*

Response: Thank you for your suggestion. The previous imaging results were problematic due to several identified factors. Firstly, we discovered that the weak specificity of the GFP antibody resulted in non-specific staining, leading to inaccurate results. Secondly, there was a decrease in the number of viable cells available for analysis during the flow cytometry process (~4h), possibly due to cell death or lysis. To overcome these challenges, we conducted additional experiments. Specifically, we directly employed isolated blood for the immunofluorescence assay in order to minimize cell death and enhance the quality of the experimental procedure. Additionally, we obtained *Salmonella*-specific antibodies (abcam, ab35156) for staining. These interventions proved successful in obtaining clearer imaging of F4/80-positive macrophages containing *Salmonella*. Importantly, each observed cell demonstrated the presence of 1-2 bacteria, as depicted in **Figure 2**. This improved imaging provides a more precise and reliable representation of the experimental findings. Consequently, we have included this figure as part of the revised version, specifically in **Figure 1H**.

Figure 2. F4/80-positive macrophages containing *Salmonella* were isolated from mouse blood after a 1-hour infection.

4. The finding that migration and invasion of Matrigel are completely blocked by the actin depolymerizing agent latrunculin B (Fig. 2F) is not surprising. Also, the images shown in Fig. 2H are too small; bacteria are barely visible.

Response: Thank you for your suggestion. We have included amplified representative images, which clearly display intracellular *Salmonella* in the new version (**Figure 3**).

Figure 3. Representative images of *Salmonella* induced actin rearrangement

The quantitation shown in Fig. 2I represents actin fluorescence intensity, yet the changes in actin organization are impossible to discern in the corresponding images.

Response: It is possible that a detailed description of the statistical method was not provided, however, it should be noted that the calculation of actin intensity were involved averaging the area within host cells. The raw data and data processing procedures have been uploaded to Mendeley Data (DOI:10.17632/t7k6g6sns3.2, Figure 2 and Figure 3). **Figure 4** illustrates representative picture of processed data. Further elaboration on the method can be found in the corresponding section (line

907-912).

	Area	Mean	Mode	Min	Max	IntDen	RawIntDen
1	0.207	27.199	21	1	92	5.618	505657
2	0.179	15.636	10	0	92	2.805	252422
3	0.12	19.346	12	0	92	2.324	209189
4	0.101	24.793	13	0	92	2.515	226314
5	0.07	25.296	17	0	92	1.759	158276
6	0.134	16.819	13	0	90	2.25	202523
7	0.13	20.395	16	0	92	2.651	238601

	Area	Mean	Mode	Min	Max	IntDen	RawIntDen
1	0.148	10.405	0	0	92	1.543	138880
2	0.134	8.867	0	0	92	1.19	107112
3	0.154	5.083	0	0	86	0.753	70497
4	0.071	7.323	0	0	85	0.518	46626
5	0.096	7.724	0	0	92	0.745	67061
6	0.129	7.434	0	0	91	0.96	86401
7	0.081	5.533	0	0	76	0.447	40236
8	0.118	7.598	0	0	92	0.899	80929
9	0.063	11.231	4	0	92	0.709	63805
10	0.127	6.143	0	0	92	0.779	70095
11	0.063	5.643	2	0	76	0.355	31929
12	0.067	5.485	0	0	92	0.367	33021
13	0.089	4.906	0	0	92	0.439	39504
14	0.135	10.103	0	0	92	1.366	122970
15	0.089	8.868	0	0	92	0.788	70925
16	0.091	4.84	0	0	92	0.441	39663
17	0.081	7.359	0	0	92	0.598	53797
18	0.097	7.021	2	0	92	0.678	61018

Figure 4. Some original graphs of statistical data for normalization of actin intensity.

The authors claim that infected cells form podosomes, but these are not detectable in the images shown. How do the authors define podosomes? What other markers of podosomes were used?

Response: Thank you for your question. Due to the fact that podosomes are not the focus of this study, we have removed the description of "infected cell forming podosomes" in the new version. I kindly request your understanding.

5. In Fig. 3 the authors generated stable macrophage cell lines (RAW264.7) expressing either WT SteC or a catalytically inactive mutant (K256H). Although the data indicate a strong dependence of motility and invasion on SteC catalytic activity, the authors need to demonstrate that the WT and mutant proteins are expressed at equivalent levels in the two lines.

Response: Thank you for your suggestion. We have used the SteC antibody (Dia-An Biotech, Custom made) to demonstrate that there is no difference in the expression of the SteC protein between the wild type and mutant strains (**Figure 5**). We have included these results in **Figure 3A** of the revised manuscript.

Figure 5. protein validation of SteC expression in WT and K256H-stable cell lines

6. *The concentrations of the inhibitors used in Fig. 3F-I should be provided.*

Response: Thank you for your suggestion. In the revised version, we have included a description of the concentration of inhibitors employed in our study (Line 870-871).

7. *The colocalization of SteC with Myl12a (Fig. 4D) needs to be quantified across at least 20 cells. What do the authors think the highlighted puncta represent? If they think these are podosomes, a corresponding podosome marker should be included in the images.*

Response: Thank you for your suggestion. In the revised version, we have substituted the representative images with new ones. The colocalization of SteC with Myl12a were quantified across 20 cells as shown in **Figure 6**, the Pearson correlation coefficient (PCC) of each cells were determined.

Figure 6. Co-localization analysis of SteC and Myl12a.

8. Fig. 5 shows convincingly that SteC phosphorylates Myl12a in vitro and in cells overexpressing SteC. Is Myl12a phosphorylated in cells infected with *S. Typhimurium*?

Response: Thank you for your valuable suggestion. Consequently, we conducted separate infection experiments using RAW264.7 and BMDM cells to assess the phosphorylation levels of Myl12a during *Salmonella* infection. Remarkably, phosphorylation of Myl12a was detected as early as 15 minutes after wild-type *Salmonella* infection, whereas no corresponding phosphorylation was observed in the Δ SteC infected cells (**Figure 7**). These findings strongly indicate that *Salmonella* activates SteC-dependent phosphorylation of Myl12a during the course of actual infection. The above data has been added to **Figure 5** of the revised manuscript.

Figure 7. *Salmonella* infection promotes phosphorylation of Myl12a in a SteC-dependent manner.

9. A table containing the results of the phosphoproteomic screen should be provided in support of Fig. 5H.

Response: As per your request, we have included the proteomic data of protein phosphorylation modification in the appendix of the article (Appendix Table.S1).

10. As noted above, the authors conclusion that SteC is required for breaching the intestinal vascular barrier is an over-interpretation of the data. The images shown in Fig. 7G and 7H have several problems. First, they are too small, and bacteria are barely detectable. Second, an earlier time point should be shown at which both WT and Δ SteC bacteria are detectable in the lamina propria. Third, images acquired from multiple sections from at least 3 different mice need to be quantified to be convincing.

Response: Thank you for your valuable suggestion. Based on your suggestions, we conducted animal infection experiments again, this time selecting 30 minutes and 1 hour after infection as time points for intestinal pathological observation. As shown in the **Figure 8 A-F**, at 30 minutes after infection, the majority of *Salmonella* were located within the intestinal epithelial cells, but a small portion of bacteria had already reached the macrophages around the blood vessels. **At 1 hour after infection, more than half of bacteria were found in the macrophages surrounding the blood vessels.** The $\Delta steC$ strains were also enriched in macrophages around the blood vessels 1 hour post infection, accounting for approximately 30% of the bacteria present in these macrophages, suggesting that the $\Delta steC$ mutant did not exhibit a failure to migrate away from infected epithelial cells. Pathological analysis of infection results also revealed that, 20 hours post oral gavage infection, $\Delta steC$ were frequently detected in cells surrounding blood vessels in intestinal tissue, whereas such occurrences were barely observed in wild-type cases (**Figure 8I, J**). We conducted the experiments using a 5 mice/group and performed statistical and quantitative analysis on the experimental results.

Figure 8. Immunofluorescence analysis revealed the precise localization of *Salmonella* during both early and late stages of infection.

A-B.Small intestinal (SI) after 0.5h (A) or 1h (B) of wild-type *Salmonella* infection

were stained with PV-1 (Red), and *Salmonella* (Green). C. Statistics of *Salmonella* distribution in host small intestinal after infection. D-E. Small intestinal (SI) after 0.5h (D) or 1h (E) of $\Delta steC$ *Salmonella* infection were stained with PV-1 (Red), and *Salmonella* (Green). F. Statistics of *Salmonella* distribution in host small intestinal after 1h infection. G. Intestinal macrophages primarily reside around the small intestinal villous microvasculature. H. *Salmonella* primarily colonizes macrophages located in the vicinity of blood vessels after 1h of *Salmonella* infection. I-J. After 20 h of infection, $\Delta steC$ still remained within the vascular intestinal macrophages.

11. Finally, there is no evidence in these images that $\Delta SteC$ bacteria are incapable of penetrating the vasculature. Their presence in the lamina propria at this single time point could easily represent a failure to migrate away from infected epithelial cells rather than an inability to penetrate the vascular basement membrane and endothelial layer.

Response: Thank you for your question. In the previous version, we indeed did not provide sufficient evidence to demonstrate that *Salmonella* SteC induces macrophage translocation across the blood vessel barrier. To further confirm this issue, we conducted new experimental explorations. Firstly, we performed immunofluorescence observations on the *Salmonella*-infected mouse intestines at early stages (30 minutes and 1 hour) of infection. By using *Salmonella*-specific antibodies instead of GFP antibodies, we achieved excellent staining results (**Figure 8**). The experimental results clearly showed the presence of *Salmonella* in the macrophages surrounding the blood vessels. **We attempted real-time live imaging to capture dynamic videos of bacteria-containing macrophages translocating through blood vessels, but we encountered multiple failures.**

To further confirm the ability of *Salmonella*-containing macrophages to translocate across the blood vessel barrier, we conducted an *in vitro* experiment simulating vascular penetration. We immobilized vascular endothelial cells in a chamber and observed the penetration of these cells by macrophages (**Figure 9**). The results indicated that uninfected macrophages had no ability to penetrate the blood vessels. However, macrophages infected with wild-type *Salmonella* exhibited a strong ability to traverse the vascular endothelial cells, while the ability of $\Delta steC$ knockout bacteria to penetrate the vascular endothelial cells was significantly reduced compared to the wild-type. Furthermore, experiments using stable SteC-expressing cells also demonstrated that cells expressing SteC rapidly traversed the vascular endothelial cells (**Figure 9C**). All the above data support our hypothesis that *Salmonella* drives macrophage translocation across the blood vessel barrier into the bloodstream.

Figure 9. *In vitro* vascular penetration assay showed that SteC drives macrophage translocation across the blood vessel barrier

12. In their discussion the authors offhandedly refer to an observation that Δ SteC strains exhibit a higher toxicity relative to WT strains. What do they mean by "toxicity" and in what context? What kind of cells?

Response: Thank you for your question. Although the spread of Δ steC in the host is weaker compared to the wild-type strain, we observed that mice infected with Δ steC strain had a shorter survival time compared to those infected with wild-type *Salmonella* (Figure 10, unpublished data). In the context of the article, "toxicity" refers to the toxic effects on animals. A detailed study on the early death of animals caused by Δ steC will be reported in a separate article.

Figure 10. A shorter survival time following Δ steC infection than WT

Do the $\Delta SteC$ strains kill their host macrophages?

We previously compared the cytotoxicity of wild-type and $\Delta steC$ on Raw264.7 cells using the LDH method, and found that there was no significant difference between them (**Figure11**).

Figure 11. Cytotoxicity of WT and $\Delta steC$ infected Raw264.7 cells

Dear Bingqing,

Thank you for submitting a revised version of your manuscript. Your study has now been seen by one of the original referees, who finds that their previous concerns have been addressed. There now remain only a few editorial points that need to be addressed before I can extend acceptance of the manuscript:

1. Please submit up to five keywords.
2. Please add the full funding information in our online system and check that it is correct and identical both in the manuscript and our online system.
3. Please add your institutional email address in our system - we require this for all corresponding authors.
4. Please add the author Wenqi Qi in our online system.
5. Please check that the funding information is correct and identical both in the manuscript and our online system.
6. Please submit a complete author checklist, which you can download from our author guidelines (<https://www.embopress.org/pb-assets/embo-site/EMBO%20Press%20Author%20Checklist-1642513524327.xlsx>). Please insert information in the checklist that is also reflected in the manuscript. The completed author checklist will also be part of the Review Process File.
7. Please rename "Summary" section into "Abstract".
8. Please upload the main and EV figures as individual production quality figure files in the .eps, .tif, or .jpg format (one file per figure). Figure legends should be moved after References.
9. Please remove Appendix information from the Data Availability section.
10. At EMBO Press we ask authors to provide source data for the main and EV figures. Our source data coordinator will contact you to discuss which figure panels we would need source data for and will also provide you with helpful tips on how to upload and organize the files.
11. During our standard figure check, we noticed potential aberrations in the Appendix Figure S4C. Please submit unmodified source data for this figure.
12. CRediT has replaced the traditional author contributions section because it offers a systematic, machine-readable author contributions format that allows for more effective research assessment. Please remove the Authors Contributions from the manuscript and use the free text boxes beneath each contributing author's name in our online submission system to add specific details on the author's contribution. More information is available in our guide to authors.
13. Please rename "Conflict of interest" section into "Disclosure and competing interests statement" (further info: <https://www.embopress.org/page/journal/14602075/authorguide#conflictsofinterest>).
14. Movies appear fully black, without a signal - please check that the files are correct. There are only two movie files uploaded, while the Appendix refers to three movies. Please check and correct.
15. Movie files need to be renamed Movie EV1 and Movie EV2; their callouts need to be updated and their legends should be zipped with the corresponding movie file. Please remove the reference to movies from the Appendix Table of Contents.
16. In the Data Availability section, please add a resolvable link for 8JBI and PXD042038 datasets. More information about the format of this section can be found here: <https://www.embopress.org/page/journal/14602075/authorguide#dataavailability>.
17. Our data editors have flagged the following issues in figure legends that need correcting:
 1. Please indicate the statistical test used for data analysis in the legends of figures 5l.
 2. Please note that in figures 7a, e; there is a mismatch between the annotated p values in the figure legend and the annotated p values in the figure file that should be corrected.
 3. Please note that information related to n is missing in the legends of figures 7a, d-e, i; k.
 4. Although 'n' is provided, please describe the nature of entity for 'n' in the legend of figure 1i.
 5. Please note that the error bars are not defined in the legends of figures 2h; 4e.
 6. Please note that the scale bar is missing for figures 7b.
 7. Please note that scale bar and its definition are missing for figures 2a; 3b; 7h, l.
 8. Please note that the white dotted circles are not defined in the legend of figure 1d. This needs to be rectified.

Finally, papers published in The EMBO Journal are accompanied online by a 'Synopsis' to enhance discoverability of the manuscript. It consists of A) a short (1-2 sentences) summary of the findings and their significance, B) 3-4 bullet points highlighting key results and C) a synopsis image that is 550x300-600 pixels large (width x height, jpeg or png format). You can either show a model or key data in the synopsis image. Please note that the image size is rather small and that text needs to be readable at the final size. Please send us this information together with the revised manuscript.

With best wishes,

leva

leva Gailite, PhD
Senior Scientific Editor
The EMBO Journal
Meyerhofstrasse 1
D-69117 Heidelberg
Tel: +4962218891309
i.gailite@embojournal.org

We realize that it is difficult to revise to a specific deadline. In the interest of protecting the conceptual advance provided by the work, we recommend a revision within 3 months (21st May 2024). Please discuss the revision progress ahead of this time with the editor if you require more time to complete the revisions. Use the link below to submit your revision:

Referee #2:

This revised manuscript is significantly improved over the original. The authors have addressed almost all of the earlier concerns by both improving on previous data and adding new data that strengthen their conclusions. This reviewer is now convinced that SteC promotes macrophage migration/invasion at least in part by phosphorylating one or more myosin light chains.

All editorial and formatting issues were resolved by the authors.

Dear Bingqing,

Thank you for addressing the final issues. I am now pleased to inform you that your manuscript has been accepted for publication in the EMBO Journal.

I will look into the synopsis text in the next couple of days and let you know if any edits to the journal style are needed.

Finally, we would like to promote your manuscript among the Chinese readership. Therefore, we would like to invite you to prepare a short summary of the manuscript in Chinese (1500-2000 Chinese characters), which we will promote on the WeChat platform 'BioArt' with more than 610,000 followers.

If you are interested in this opportunity, we recommend covering the article very close to its online publication date. Thus, ideally we would very much appreciate if you could send us a draft within the next 7 working days. Please let us know whether or not you would be interested in contributing such a short summary in Chinese.

I have included below some general guidelines on how to prepare a summary and a link to recent examples for your reference. Please let me know if you have any questions about this.

If you have any questions, please do not hesitate to contact the Editorial Office. Thank you for this contribution to The EMBO Journal and congratulations on a nice study!

Best wishes,

Ieva

General WeChat Summary Guidelines

1. These summary articles are meant to be targeting general audience, so please limit the use of specialized technical terms, acronyms and jargon.
2. A summary usually starts with brief background information of the reported work, which is followed by explaining the findings in some detail, and ends with a short review of the conclusions as well as the implications of the work and future directions for the research.
3. The summary should at least contain one graphical item, such as a scheme or a figure from the paper.
4. Please provide ONE SINGLE document containing all text and graphical materials, ideally as a Word.docx or .doc file. Please DO NOT provide the document as a .pdf file.
5. Please DO NOT publicly release the document before the paper is officially published online.

Summary Examples

EMBO J | 罗招庆/欧阳松应揭示谷酰胺脱氨酶MvcA的去泛素化功能

EMBO J | 王松灵院士团队揭示组织内应力调控大型哺乳动物乳恒牙替换的新机制
